# Preclinical and randomized phase I studies of plitidepsin in adults hospitalized with COVID-19

Jose F Varona[1,2], Pedro Landete[3,4], Jose A Lopez-Martin[5], Vicente Estrada[6,7], Roger Paredes[8,9], Pablo Guisado-Vasco[10,11], Lucia Fernandez de Orueta[11,12], Miguel Torralba[13,14], Jesus Fortun[15], Roberto Vates[12], Jose Barberan[1,2], Bonaventura Clotet[8,9,16,17], Julio Ancochea[3,4,18], Daniel Carnevali[10,11], Noemi Cabello[19], Lourdes Porras[20], Paloma Gijon[21], Alfonso Monereo[12], Daniel Abad[11,12], Sonia Zuñiga[22], Isabel Sola[22], Jordi Rodon[23], Julia Vergara-Alert[23], Nuria Izquierdo-Useros[24,25], Salvador Fudio[26], Maria Jose Pontes[27], Beatriz de Rivas[27], Patricia Giron de Velasco[5], Antonio Nieto[28], Javier Gomez[28], Pablo Aviles[29], Rubin Lubomirov[26], Alvaro Belgrano[28], Belen Sopesen[5,30,31], Kris M White[32,33], Romel Rosales[32,33], Soner Yildiz[32,33], Ann-Kathrin Reuschl[34], Lucy G Thorne[34], Clare Jolly[34], Greg J Towers[34], Lorena Zuliani-Alvarez[35,36,37,38], Mehdi Bouhaddou[35,36,37,38], Kirsten Obernier[35,36,37,38], Briana L McGovern[32,33], M Luis Rodriguez[32,33], Luis Enjuanes[22], Jose M Fernandez-Sousa[39], Nevan J Krogan[32,35,36,37,38], Jose M Jimeno[5,*], Adolfo Garcia-Sastre[32,33,40,41,*]

Plitidepsin, a marine-derived cyclic-peptide, inhibits SARS-CoV-2 replication at nanomolar concentrations by targeting the host protein eukaryotic translation elongation factor 1A. Here, we show that plitidepsin distributes preferentially to lung over plasma, with similar potency against across several SARS-CoV-2 variants in preclinical studies. Simultaneously, in this randomized, parallel, open-label, proof-of-concept study (NCT04382066) conducted in 10 Spanish hospitals between May and November 2020, 46 adult hospitalized patients with confirmed SARS-CoV-2 infection received either 1.5 mg (n = 15), 2.0 mg (n = 16), or 2.5 mg (n = 15) plitidepsin once daily for 3 d. The primary objective was safety; viral load kinetics, mortality, need for increased respiratory support, and dose selection were secondary end points. One patient withdrew consent before starting procedures; 45 initiated treatment; one withdrew because of hypersensitivity. Two Grade 3 treatment-related adverse events were observed (hypersensitivity and diarrhea). Treatment-related adverse events affecting more than 5% of patients were nausea (42.2%), vomiting (15.6%), and diarrhea (6.7%). Mean viral load reductions from baseline were 1.35, 2.35, 3.25, and 3.85 $\log_{10}$ at days 4, 7, 15, and 31. Nonmechanical invasive ventilation was required in 8 of 44 evaluable patients (16.0%); six patients required intensive care support (13.6%), and three patients (6.7%) died (COVID-19–related). Plitidepsin has a favorable safety profile in patients with COVID-19.

## Introduction

As of December 2021, there have been more than 269 million confirmed cases of coronavirus disease (COVID-19) reported to the

[1]Departamento de Medicina Interna, Hospital Universitario HM Monteprincipe, HM Hospitales, Madrid, Spain   [2]Facultad de Medicina, Universidad San Pablo-CEU, Madrid, Spain   [3]Hospital Universitario La Princesa, Madrid, Spain   [4]Universidad Autónoma de Madrid, Madrid, Spain   [5]Virology and Inflammation Unit, PharmaMar, SA, Madrid, Spain   [6]Hospital Clínico San Carlos, Madrid, Spain   [7]Universidad Complutense de Madrid, Madrid, Spain   [8]Infectious Diseases Department, IrsiCaixa AIDS Research Institute, Barcelona, Spain   [9]Hospital Germans Trias I Pujol, Barcelona, Spain   [10]Hospital Universitario Quironsalud Madrid, Madrid, Spain   [11]Universidad Europea, Madrid, Spain   [12]Internal Medicine Department, Hospital Universitario de Getafe, Madrid, Spain   [13]Health Sciences Faculty, University of Alcalá, Madrid, Spain   [14]Guadalajara University Hospital, Guadalajara, Spain   [15]Hospital Universitario Ramón y Cajal, Madrid, Spain   [16]Universitat Autònoma de Barcelona, Barcelona, Spain   [17]Universitat de Vic, Universitat Central de Catalunya, Barcelona, Spain   [18]Centro de Investigación en Red de Enfermedades Respiratorias (CIBERES), Instituto de Salud Carlos III (ISCIII), Madrid, Spain   [19]Infectious Diseases Department, Clinico San Carlos University Hospital, Madrid, Spain   [20]Internal Medicine, Hospital General de Ciudad Real, Ciudad Real, Spain   [21]Clinical Microbiology and Infectious Diseases Department, Hospital General Universitario Gregorio Marañón, Instituto de Investigación Sanitaria Gregorio Marañón, Madrid, Spain   [22]Department of Molecular and Cell Biology, Centro Nacional de Biotecnología (CNB-CSIC), Madrid, Spain   [23]IRTA, Centre de Recerca en Sanitat Animal (CReSA, IRTA-UAB), Campus de la UAB, Bellaterra, Spain   [24]IrsiCaixa AIDS Research Institute, Barcelona, Spain   [25]Germans Trias I Pujol Research Institute (IGTP), Badalona, Spain   [26]Clinical Pharmacology Unit, PharmaMar, Madrid, Spain   [27]Medical Affairs, PharmaMar, Madrid, Spain   [28]Bio Statistics Unit, PharmaMar, Madrid, Spain   [29]Preclinical Unit, Pharmamar, Madrid, Spain   [30]Sylentis, SAU, Madrid, Spain   [31]Biocross, SL, Valladolid, Spain   [32]Department of Microbiology, Icahn School of Medicine at Mount Sinai, New York, NY, USA   [33]Global Health Emerging Pathogens Institute, Icahn School of Medicine at Mount Sinai, New York, NY, USA   [34]Division of Infection and Immunity, University College London, London, UK   [35]Quantitative Biosciences Institute (QBI), University of California San Francisco, San Francisco, CA, USA   [36]J David Gladstone Institutes, San Francisco, CA, USA   [37]QBI, Coronavirus Research Group (QCRG), San Francisco, CA, USA   [38]Department of Cellular and Molecular Pharmacology, University of California, San Francisco, San Francisco, CA, USA   [39]Pharmamar, Madrid, Spain   [40]Department of Medicine, Division of Infectious Diseases, Icahn School of Medicine at Mount Sinai, New York, NY, USA   [41]Tish Cancer Institute, Icahn School of Medicine at Mount Sinai, New York, NY, USA

Correspondence: jfvarona@hmhospitales.com
*José M Jimeno and Adolfo García-Sastre contributed equally to this work.

World Health Organization, including over 5 million deaths ([1]). More than 1 yr after being officially declared a pandemic, substantial disease burden remains, as the clinical course from severe lung involvement, evolving to respiratory failure continues to be the main cause of death for COVID-19 patients. The lack of effective antiviral therapies represents a glaring unmet need, not only for the treatment of the current severe acute respiratory syndrome coronavirus 2 (SARS-CoV-2) pandemic ([2]), but also for potential future pandemics, which may originate from other emergent coronaviruses ([3], [4], [5]).

SARS-CoV-2 is a spherical, enveloped virus, around 80–120 nm in diameter. Within the lipid bilayer envelope, the viral single-stranded RNA genome is packaged within a protein capsid, which comprised the nucleocapsid (N) protein ([6]). The N protein is produced abundantly in infected cells and is a key element involved in packaging of the viral RNA genome ([7]). The SARS-CoV-2 N protein, as well as several proteins from other viruses, has been shown to bind directly to eukaryotic elongation factor $1\alpha$ (eEF1A), an important host factor for the replication of many viral pathogens ([7], [8]). Down-regulation via small interfering RNA or chemical inhibition of eEF1A has been shown to result in a significant reduction in the replication and infectivity of several viruses, including SARS-CoV-2 ([9], [10], [11]).

Plitidepsin is a cyclic depsipeptide originally isolated from a Mediterranean marine tunicate (*Aplidium albicans*) that has been shown to interact directly and inhibit the activity of eEF1A ([12]). Experiments in cell culture and mice models have shown that plitidepsin can inhibit SARS-CoV-2 replication, indicating that plitidepsin may be a promising candidate for the treatment of COVID-19 ([8]).

Plitidepsin has undergone an extensive clinical development program for the treatment of cancer. Pharmacokinetic and safety properties of plitidepsin have been gathered from several Phase I and II clinical trials which have also explored different i.v. dosing schedules and infusion times ([13], [14], [15], [16], [17]). Based on the results obtained from a Phase III clinical trial (ADMYRE) ([18]), the Australian Therapeutic Goods Administration approved the combination of plitidepsin with dexamethasone for the treatment of patients with relapsed/refractory multiple myeloma in 2018 ([19]).

To date, no clinical trials have evaluated plitidepsin for the treatment of infectious disease. In this proof-of concept clinical trial (APLICOV-PC), we sought to determine the safety and toxicological profile of plitidepsin, as well as to explore any potential efficacy effects, across three dose levels in patients hospitalized with COVID-19.

# Results

### Plitidepsin shows potent inhibition of viral replication in vitro

The antiviral activity of plitidepsin was evaluated by three separate teams against different coronavirus species, strains and variants. Treatment of Huh-7 cells with as little as 0.5 nM of plitidepsin inhibited infection of a human coronavirus 229E expressing GFP (Fig 1A). A $10^4$-fold decrease in SARS-CoV genomic RNA accumulation and a $10^3$-fold decrease in virus SARS-CoV titers were observed in Vero E6 cells treated with 50 nM plitidepsin (Fig 1B and Table S1). By

comparison, and consistent with previous results ([8], [20]), plitidepsin showed nanomolar efficacy against SARS-CoV-2-induced cytopathic effects in Vero E6 cells with a half-maximal inhibitory concentration ($IC_{50}$) of 0.038 $\mu$M, at concentrations where no cytotoxic effects were observed ($CC_{50}$ 2.9 $\mu$M) (Fig 1C). Moreover, plitidepsin maintained its nanomolar potency against replication of early as well as later SARS-CoV-2 lineages, such as B.1.1.7 ($\alpha$), B.1.351 ($\beta$), B.1.617.2 ($\delta$), B.1.621 ($\mu$), and B.1.1.529 ($o$) variants (Fig 2A–F). Noteworthy, in human lung and gastrointestinal cell lines plitidepsin was more effective against both early and $\alpha$ variants than remdesivir (Fig 2G).

### Predicting the effective dose of plitidepsin in COVID-19

To identify target human plasma concentrations of plitidepsin for SARS-CoV-2 infection we developed an extrapolation from in vitro results, in line with current recommendations ([21]). This approach integrated results from nonclinical studies, including in vitro drug sensitivity data for SARS-CoV-2 in Vero cells, human plasma protein binding data (98%) (Table S2), and in vivo tissue distribution data in rats (lung-to-plasma partition coefficient ratio of 543-fold; Table S3).

The target plasma and lung concentrations for plitidepsin were initially based on in vitro data obtained by Boryung Pharmaceuticals, which established an $IC_{50}$ of 3.26 nM and an $IC_{90}$ of 9.38 nM. A validated pharmacokinetic population model of plitidepsin ([22]) was used to simulate plasma exposures at different dose levels and infusion durations, so that plitidepsin plasma profiles would reach 0.33, and 0.96 $\mu$g/l, assuring target concentrations in lung above the aforementioned in vitro $IC_{50}$ and $IC_{90}$. A 3-d daily schedule was initially selected to achieve sustained active exposures, under the hypothesis that an acute reduction of the viral load would prevent the onset of the more severe inflammatory phase of COVID-19. The predicted plasma concentrations of plitidepsin, at a dose of 1.5 mg infused i.v. over 90 min, were above the target $IC_{50}$ for the full treatment period and above the $IC_{90}$ for half of the treatment period. The respective predictions after a dose of 2.5 mg were above the $IC_{90}$ during most of the treatment period (Fig 3). Thus, we anticipated that the proposed range of doses would result in stable active concentrations in critical anatomical compartments, such as the lung, for more than 120 h. This model was later supported by White et al, who reported an $IC_{90}$ of 0.88 nM ([8]). To reach this target concentration in lung tissue, according to the above reasoning, the plitidepsin plasma concentration should be above 0.18 $\mu$g/l.

### Patient characteristics

In total, 46 hospitalized COVID-19 patients were enrolled across 10 sites in Spain (Fig 4). A diagram of the per-protocol treatment can be found in Fig 5, and the complete study protocol can be found in Supplemental Data 1. Baseline demographic and clinical characteristics are summarized in Table 1. The average patient age was 52 yr (range 31–84 yr). Most patients were male (66.7%) and 80% had co-morbidities (46.7% had two or more). The most commonly reported comorbidities were obesity (22.2%), hypertension (20%), and type 2 diabetes mellitus (17.8%). The distribution of comorbidities was similar among the three treatment cohorts.

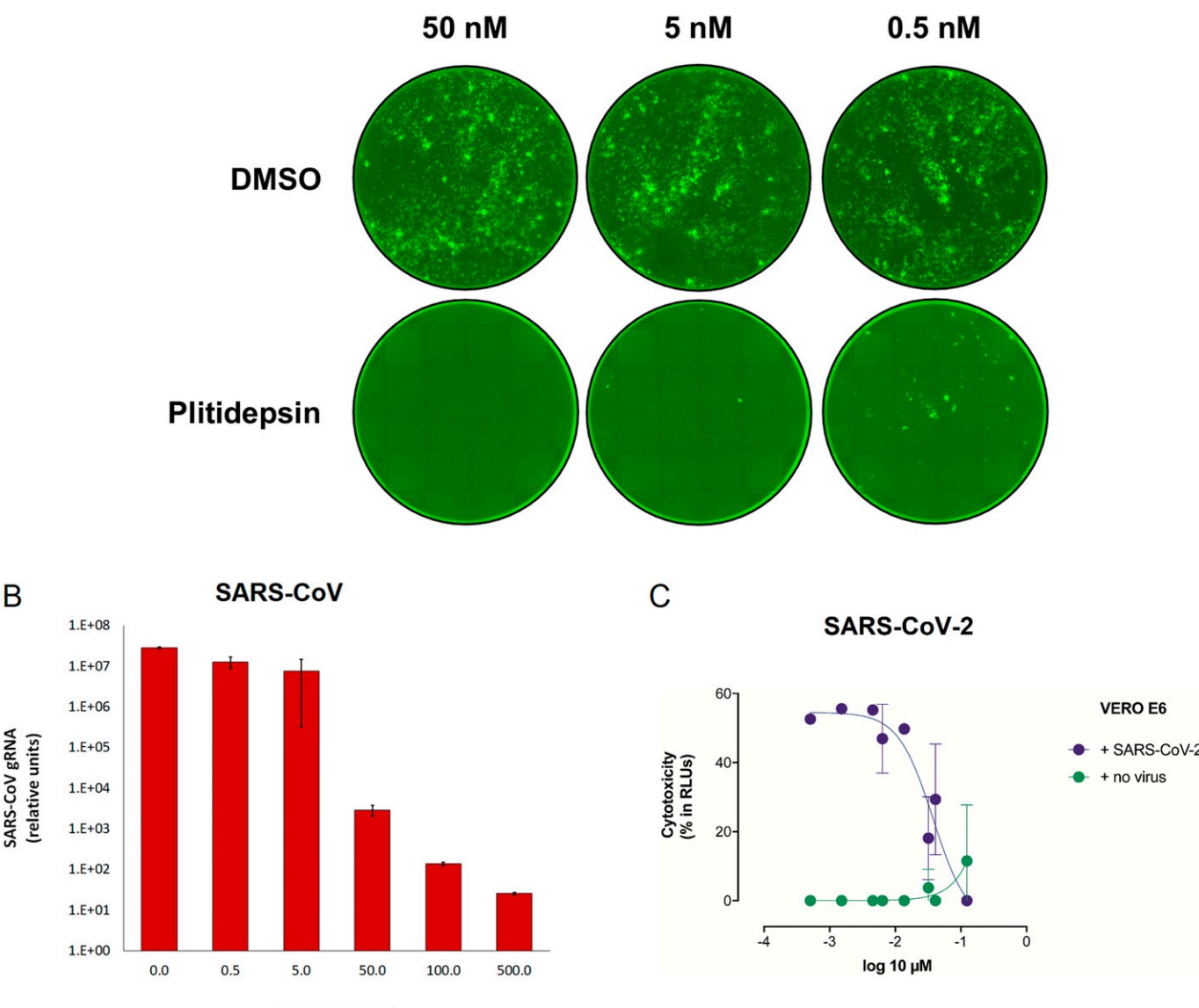

**Figure 1. Plitidepsin shows strong antiviral activity in vitro against different coronavirus species.**
**(A)** Treatment of Huh-7 cells with 0.5–50 nM of plitidepsin inhibited infection of a human coronavirus 229E expressing green fluorescent protein. All cells were treated 8 h after infection and fluorescent foci were analyzed at 48 h. **(B)** Accumulation of SARS-CoV genomic RNA is inhibited with increasing doses of plitidepsin. Confluent Vero E6 cells were infected with SARS-CoV and subsequently treated with plitidepsin at varying concentrations 1 hour post infection. Viral genomic RNA was measured 48 hours post infection. **(C)** Cytopathic effect on Vero E6 cells exposed to a fixed concentration of SARS-CoV-2 in the presence of increasing concentrations of plitidepsin. Plitidepsin was used at a concentration ranging from 5 nM to 100 $\mu$M. Nonlinear fit to a variable response curve from one representative experiment with two replicates is shown (blue), excluding data from drug concentrations with associated toxicity; cytotoxicity in the absence of virus is also shown (green). Error bars represent SD; points without error bars have a SD that is too small to visualize. DMSO, dimethyl sulfoxide; RLU, relative light unit.

Most patients had moderate COVID-19 (51.1%), according to the US Food and Drug Administration (FDA) categorization (23), with 13.3% and 35.6% having mild and severe disease, respectively. Baseline chest X-rays showed evidence of lower respiratory infection (infiltrates, unilateral pneumonia, or bilateral pneumonia) in 41 of 45 treated patients (91%), with bilateral pneumonia seen in 32 of them (71%); the percentage of patients with bilateral pneumonia was similar across dose cohorts. Viral load was similar across the three cohorts, with average baseline values for SARS-CoV-2 RNA from nasopharyngeal samples of 6.1 $\log_{10}$ copies/ml as measured by quantitative (q)RT-PCR.

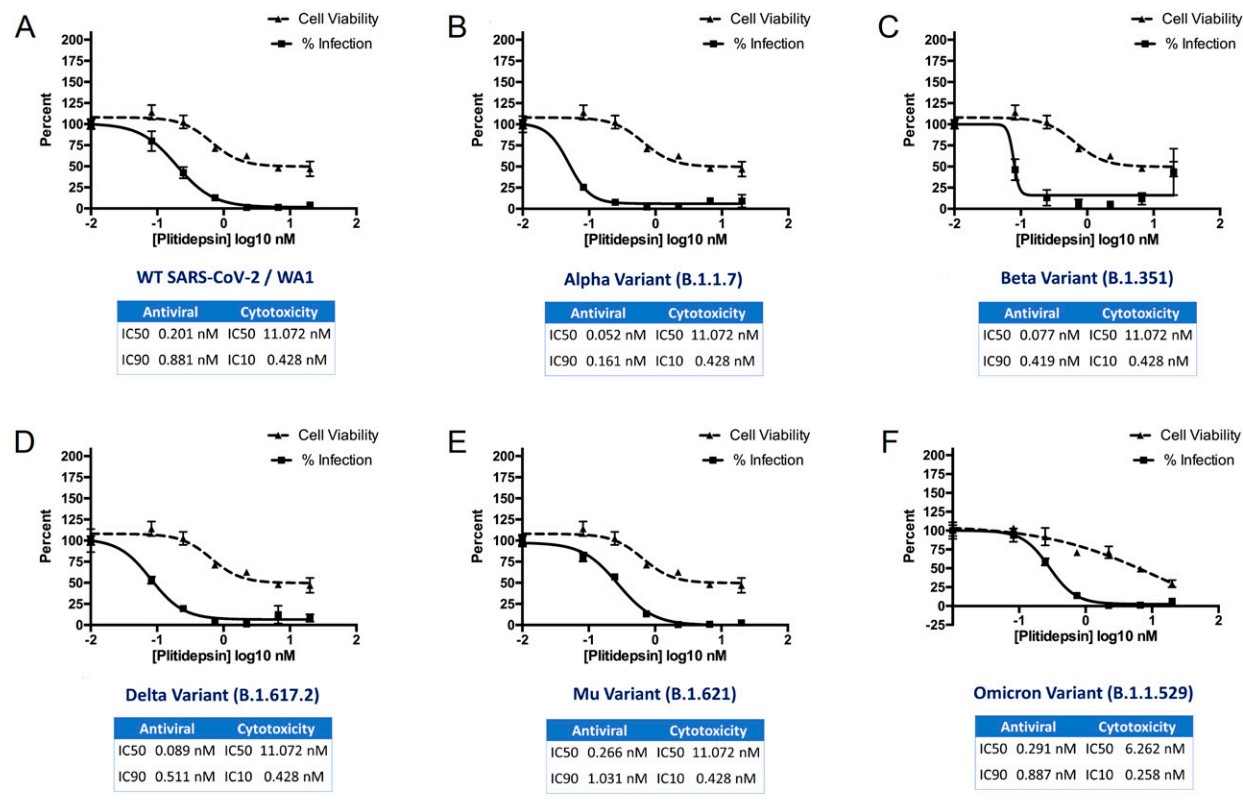

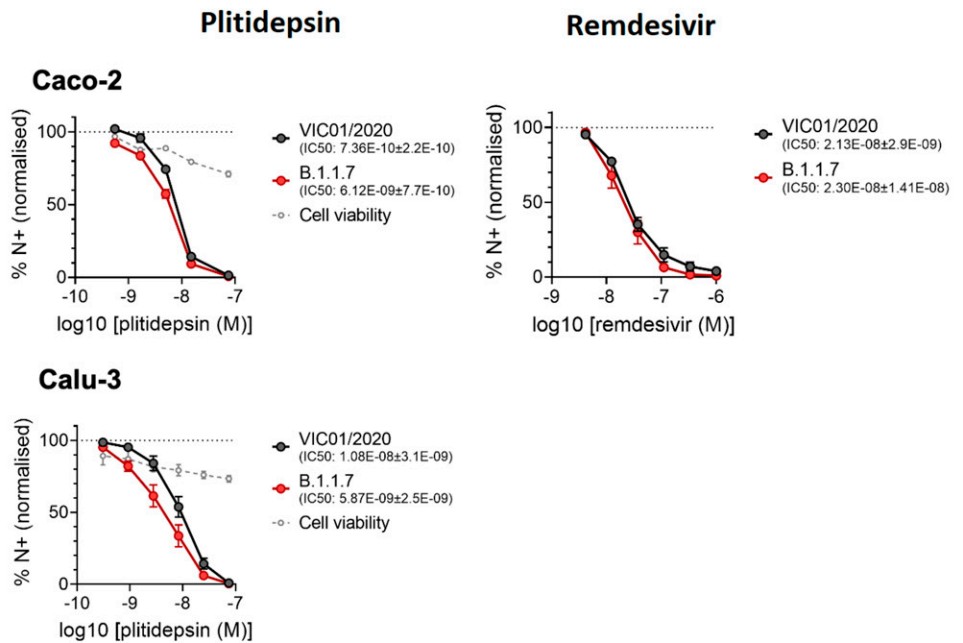

**Figure 2. Plitidepsin shows strong antiviral activity in vitro against SARS-CoV-2 variants.**
**(A, B, C, D, E, F)** Plitidepsin inhibits SARS-CoV-2 variants. HeLa-ACE2 cells were pretreated with plitidepsin or DMSO control 2 h after infection with (A) SARS-CoV-2/WA1, (B) α (B.1.1.7), (C) β (B.1.351), (D) δ (B.1.617.2), (E) μ (B.1.621), or (F) o (B.1.1.529). Virus infectivity was measured 48 h postinfection. Cytotoxicity was performed in uninfected HeLa-ACE2 cells with same compound dilutions and concurrent with viral replication assay. Error bars represent SD across biologically independent triplicates. **(G)** Plitidepsin efficacy against early and α (B.1.1.7) variants compared to remdesivir. Calu-3 and Caco-2 cells were pre-treated with plitidepsin, remdesivir, or DMSO control at the indicated concentrations at an equivalent dilution for 2 h before SARS-CoV-2 infection. Cells were harvested after 24 h for analysis, and viral infection measured by intracellular detection of SARS-CoV-2 nucleoprotein by flow cytometry. Tetrazolium salt (MTT) assay was performed to verify cell viability. Error bars represent standard error of the mean. $IC_{50}$: half maximal inhibitory concentration.

## Plitidepsin treatment was generally well tolerated in hospitalized patients with COVID-19

Study interventions are described in Fig 5, the Materials and Methods section, and Supplemental Data 1 (protocol). One patient withdrew consent before initiating any study-specific procedure. Forty-four patients completed the study through day 31. One patient in the 1.5-mg cohort withdrew from the study before completing the full treatment because of a grade 3 hypersensitivity reaction occurring shortly after the initiation of the first infusion of plitidepsin. This happened despite the pre-treatment with oral dexamethasone 8 mg. The study protocol was thereafter amended to require IV premedication with dexamethasone phosphate 8 mg, instead of oral administration, as well as IV ondansetron 8 mg, followed by 4 mg orally every 12 h until 48 h after the last administration of plitidepsin (previously this was left to physician's discretion) (Fig 5; see Supplemental Data 2 for a summary of all study amendments).

All 45 of the treated patients were evaluable for safety. Three patients in this study died (6.7%); all had severe disease at baseline, and each death was determined to be related to COVID-19. Deaths occurred on days 22, 30, and 57 after the start of treatment with plitidepsin. One patient received plitidepsin 1.5 mg/day and the other two received 2.5 mg/day, with no reported tolerability issues.

Seven additional patients experienced serious adverse events: five dosed at 1.5 mg/day, 1 dosed at 2.0 mg/day, and 1 dosed at 2.5 mg/d. As previously mentioned, only one serious adverse event (2.2% subjects) was considered related to the study drug: a Grade 3 hypersensitivity reaction occurring ~5 min after the start of the first infusion of plitidepsin.

Although nearly all (44 of 45; 97.8%) patients experienced one or more adverse events (AEs), they were determined to be treatment related in only 25 patients (55.5%). Regardless of causality, 14 (31%) patients experienced at least one Grade ≥ 3 AE according to National Cancer Institute–Common Toxicity Criteria for AEs, version 5.0 (NCI-CTCAE v5). The prevalence of grade 3–4 AEs was 40.0% in the 2.5 mg cohort, 20.0% in the 2.0 mg cohort, and 33.3% in the 1.5 mg cohort. Although almost all Grade ≥ 3 AEs were attributed to COVID-19, two Grade 3 AEs were attributed to plitidepsin: one case each of anaphylactic reaction (at 1.5 mg/day) and diarrhea (at 2.5 mg/day). No Grade 4 AEs were reported.

Table 2 presents the frequency of treatment-related AEs in this study. The following treatment-related AEs occurred in more than one patient: nausea (42.2%), vomiting (15.6%), diarrhea (6.7%), abdominal pain (4.4%), dizziness (4.4%), and dysgeusia (4.4%). These events were all mild to moderate (Grade 1–2) except the one case of Grade 3 diarrhea described above. The implementation of the aforementioned protocol amendment was associated with a reduction in the proportion of patients with nausea (from 55.6% to 38.9%) and vomiting (from 22.2% to 13.9%). No new hypersensitivity reactions were seen in any of the 36 patients treated after the changes to premedication described above (108 infusions).

Several laboratory abnormalities were reported in these patients, most of which were consistent with the acute, inflammatory nature of COVID-19. Of note, plitidepsin did not show signs of clinically relevant hemotoxicity; there were two patients with neutropenia (one Grade 1 and another Grade 2). An isolated observation of Grade 3 neutropenia was reported in an asymptomatic outpatient during follow-up at Day 31; this patient was also taking metamizole, and the investigator responsible deemed that the event was neither clinically relevant nor related to plitidepsin. Of the 32 patients who entered with normal platelet counts, only one had Grade 1 thrombocytopenia, whereas of the 12 patients who entered into the study with Grade 1 thrombocytopenia, 5 (41.7%) had counts normalized.

Regardless of causality, abnormalities in liver function tests were common, transient, mild, and reversible. Elevation of alanine aminotransferase (ALT) and aspartate aminotransferase (AST) were reported in 29 of 44 (66%) and 13 of 44 (30%) patients, respectively. Two patients had a single and self-limited observation of a Grade 3 increase in ALT, with no associated increase in bilirubin (Fig S1). Four patients of the 42 who entered with normal creatinine developed a Grade 1 increase in creatinine on study. A Grade 1 increase in creatinine phosphokinase (CPK) was documented in 3 of 37 patients (8.1%) who had normal baseline values, whereas four of five patients (80%) who entered with Grade 1–2 elevation had their CPK values decreased on study.

Hyperglycemia was documented in 8 of 45 patients (18%): three patients in the 1.5 mg/day dose cohort, two patients in the 2 mg/day, and three patients in the 2.5 mg/day group. All cases of hyperglycemia were Grade ≤ 2 except one patient, who had Grade 3 hyperglycemia lasting for 2 d. For five patients, hyperglycemia was considered related to concomitant medication. None of these events were considered to be related to plitidepsin.

Finally, protocol-specified analysis of electrocardiograms (ECGs) was conducted by a third-party central laboratory (ERT, Inc.). A total of 317 ECGs from 44 patients were machine-readable and eligible for this analysis. No single value was above the reference limit values of concern regarding left ventricular repolarization (namely, no corrected QT interval by Fredericia [QTcF] was >480 ms and no δ QTcF was >60 ms) in any of the evaluable patients. No significant effects either on atrioventricular conduction or on depolarization, as measured by mean changes in PR and QRS intervals, were observed. No new clinically relevant morphological changes were observed, except for a few isolated ST segment, T wave, and conduction abnormalities, likely explained by the consequences of acute infection.

### Results on secondary efficacy end points

After treatment, patients' viral loads were evaluated by RT-PCR. Viral load showed mean declines from baseline of 1.35, 2.35, 3.25, and 3.85 $\log_{10}$ copies/ml at days 4, 7, 15, and 31, respectively (Table 3; see also Table S4 for individual assessments). The mean time to undetectable viral load was 13 d and was longer in patients with severe disease at baseline (15 d) than in those with mild or moderate disease (12 d) (Fig 6A). There were no significant differences in change in viral load across dose levels (Table 3 and Fig 6B).

Nonmechanical invasive ventilation was required in one patient with moderate disease (4.3%) and in seven patients with severe disease (46.7%). Six patients required intensive care support (13.6%), all of whom had severe disease at baseline (6 of 15, 40%). Tables 3, S5, and S6 summarize additional outcome measures. Fig S2

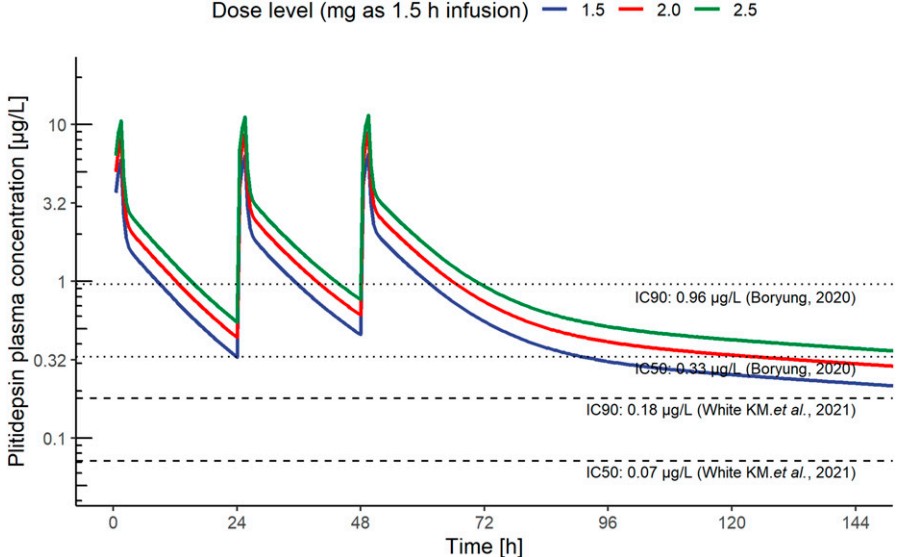

**Figure 3. Pharmacological estimation of active plasma concentrations of plitidepsin.**
Predicted plasma concentrations achieved by a 90 min i.v. infusion of plitidepsin (1.5, 2, and 2.5 mg) and plasma $IC_{50}$ and $IC_{90}$ thresholds to assure concentrations in lung above $IC_{50}$ and $IC_{90}$ established in vitro, respectively (8, 52). Results were used to support the study doses and schedule. $IC_{50}$: half maximal inhibitory concentration; $IC_{90}$: 90% of maximal inhibitory concentration.

plots the evolution of a six-category ordinal scale that integrates the need for hospitalization and oxygen therapy (24) for each participant, at prespecified time points.

While on study, 64.4% (29 of 45) of patients received systemic corticosteroids for the treatment of COVID-19 manifestations, beyond their use on days 1–3 as the per protocol pre-medication. They were similarly distributed across the dosing groups: 9 patients in the 1.5 mg/d and 10 patients each in the 2 mg/d and 2.5 mg/d, for a respective median duration of 16 d (interquartile range [IQR]: 11–32 d), 8.5 d (IQR: 5–31 d), and 17.5 d (IQR: 13–35 d). After completing plitidepsin treatment, 15.5% (7 of 45) of patients received other additional treatments for COVID-19, including the anti-viral agent remdesivir (one patient) and/or the anti-IL-6 monoclonal antibody tocilizumab (six patients).

### Post hoc analysis of hospital discharge rates

All 44 patients who completed the 3-d treatment with plitidepsin were evaluated for efficacy analyses. Overall, the discharge rates by days 8 and 15 after the start of plitidepsin were 56.8% (25 of 44) and 81.8% (36 of 44), respectively. Without adjusting for any covariates and with the constraints of the small sample size, there was no clear dose effect on the time to hospital discharge (Fig 7A and B) and on discharge rates at Day 15: 78.6% (1.5 mg/day), 93.3% (2 mg/day), and 73.3% (2.5 mg/day). Nevertheless, the proportion of patients achieving hospital discharge by Day 8 seemed to increase with dose, from 42.9% for those receiving 1.5 mg/day to 60% and 66.7% for those receiving 2 and 2.5 mg/day, respectively.

The median time to discharge was 7 d (IQR: 7–9 d) in patients with mild COVID-19, 7 d (IQR: 6–8 d) with moderate COVID-19, and 14 d (IQR: 7–26 d) with severe COVID-19 at baseline (overall log-rank *P* = 0.001) (Figs S3 and S4 and Table S6). Figs 7B and S4 plot the length of the hospitalization and time in the intensive care unit per subject, dose, and severity of the disease. As expected, the length of hospitalization was greater in patients with severe disease at baseline. The discharge rate by Day 15 was 95.7% (22 of 23) in

patients with moderate disease compared with 53.3% (8 of 15) in patients with severe disease, and 100% (6 of 6) in patients with mild disease. Similarly, the discharge rate by Day 8 was 73.9% (17 of 23) in patients with moderate disease compared with 27% (4 of 15) in patients with severe disease and 66.7% (4 of 6) in patients with mild disease (Table 3).

Baseline viral load was found to be significantly correlated with hospital discharge by Day 15, by logistic and Cox regression models. This occurred despite the limitations for modeling (because of the small number of patients [N = 44] and the high percentage of patients [82%] discharged by Day 15), following a stepwise selection of covariates with a *P*-value < 0.10 in univariate logistic models.

In addition to viral PCR reduction, patients showed analytical improvement of biomarkers associated to inflammatory processes, such as lymphocyte count and C-reactive protein (CRP) (Figs S5 and S6).

Because patients with moderate COVID-19 at baseline represented the largest subgroup in this study and may represent the potential target population for further development of plitidepsin as a COVID-19 therapy, we conducted an additional post hoc exploratory analysis in these patients as part of hypothesis generation for Phase III study design. This analysis showed that all patients with moderate disease at baseline who were allocated to the highest dose level of 2.5 mg/day (8 of 8) were discharged by day 8, whereas three of seven patients (43%) and three of eight patients (38%) treated at 2.0 mg/day and 1.5 mg/day, respectively, were discharged beyond that time point. These data are visualized in Fig 8A, which shows that for patients with moderate disease who received plitidepsin 2.5 mg/day, the most probable duration of hospitalization was ~1 wk. Furthermore, the variation in hospitalization duration in patients receiving 2.5 mg/day was narrower than that seen with other doses.

Although statistical analyses were not performed because of the small sample size, visual exploration of mean trends do not appear to capture dose-dependent differences in the kinetics of viral PCR in the subgroup of patients with moderate COVID-19 (data not

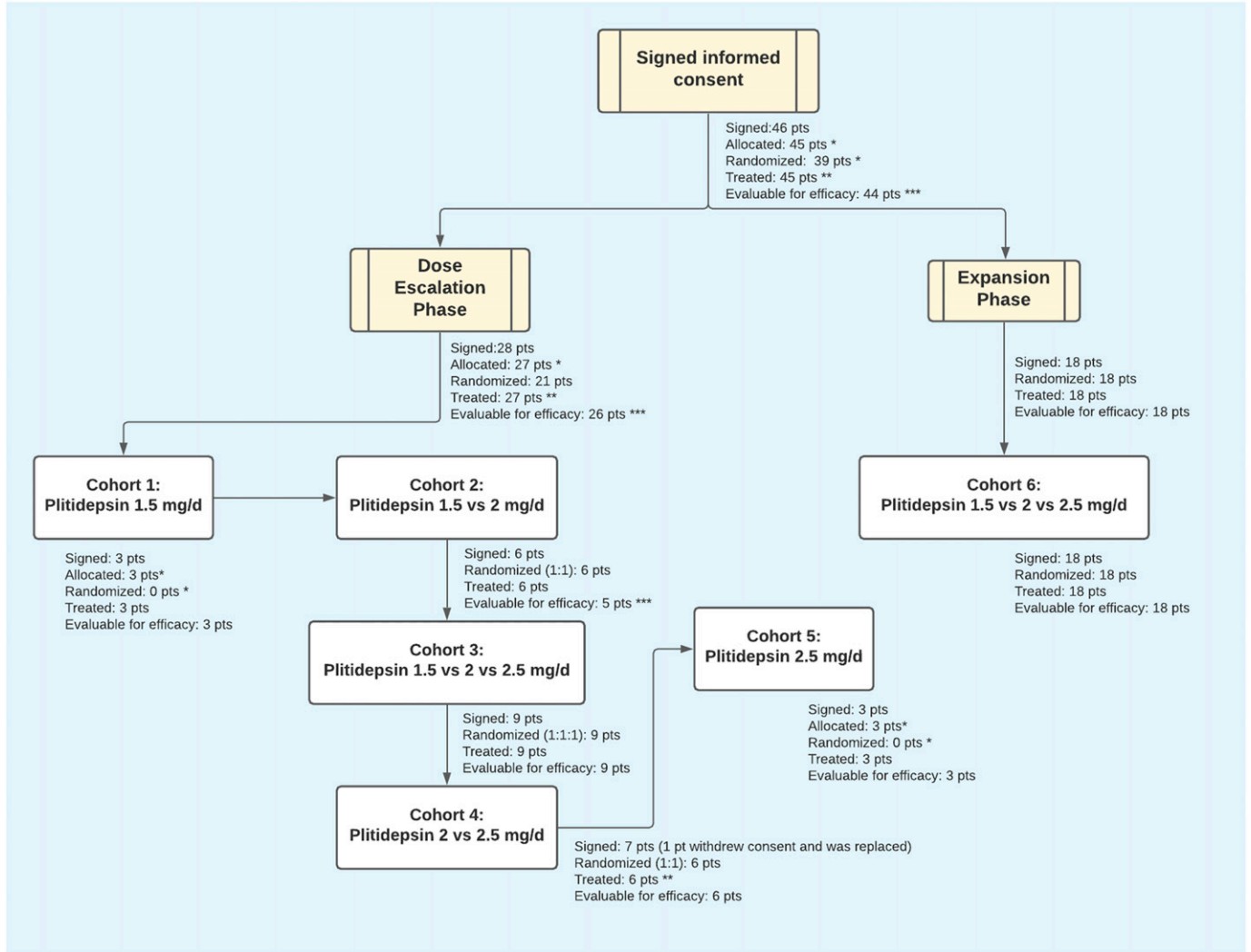

**Figure 4. Study Flow (CONSORT).**
**\*** For safety reasons, the first three patients of the study were sequentially allocated at the lowest dose level. Inclusion in the highest dose group was opened when three patients had been randomized to the intermediate dose. For that reason, the last three patients treated at the highest dose were also sequentially allocated. **\*\*** One patient withdrew consent before starting any study procedure and was replaced. **\*\*\*** All treated patients were evaluated for safety. All patients who completed treatment were assessed for efficacy. One patient experienced a grade 3 hypersensitivity reaction, shortly after the start of day 1 infusion of plitidepsin. This patient did not complete therapy, discontinued the study for safety reasons and was not evaluable for efficacy. This patient was not replaced. d, days; pts, patients.

shown). Nevertheless, it suggests that the higher the dose of plitidepsin, the faster the increase in lymphocyte count (Fig S7A), the smaller the peak of CRP at day 7 (Fig S7B), and the faster the recovery of neutrophil-to-lymphocyte ratio (NLR) (Fig S7C), D-Dimer (Fig S7D), and in the score of a six-ordinal scale for outcome (Fig 8B, Table S5, and Fig S8) (24) (see footnote on Fig 8B for description of the ordinal scale). Corresponding improvement of lung infiltrates were also observed in non-protocol chest radiographs, an example of which is shown in Fig S9A–C.

## Discussion

Despite worldwide efforts to identify new treatments, as of this publication, no highly effective antiviral therapy against SARS-

CoV-2 is yet available. Several strategies have been attempted with limited effect (25). One large collaborative effort systematically mapped the interactome between SARS-CoV-2 proteins and human proteins, identifying several dozens of potentially druggable interactions (11). Notably, the authors highlighted the potent antiviral effects after the inhibition of eEF1A, which had been previously described as the target of plitidepsin (12).

In the set of preclinical studies of this and previous reports, plitidepsin showed strong antiviral activity and a positive therapeutic index in in vitro models of SARS-CoV-2 infection, with better performance than other drugs, including remdesivir (8, 26 Preprint). In our study, regardless of the coronavirus species (HCoV 227E, SARS-CoV, and SARS-CoV-2), the host cells, or the quantifying method used, highly consistent results were obtained, with the $IC_{50}$ of plitidepsin always being in the nanomolar range. Notably, a similar in vitro

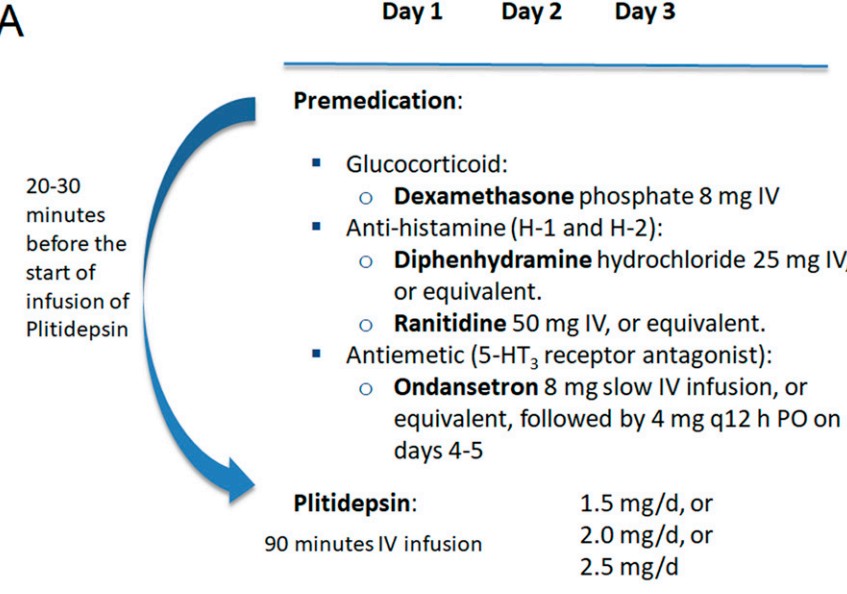

**A**

Day 1    Day 2    Day 3

**Premedication:**

20-30 minutes before the start of infusion of Plitidepsin

- Glucocorticoid:
  - ○ **Dexamethasone** phosphate 8 mg IV
- Anti-histamine (H-1 and H-2):
  - ○ **Diphenhydramine** hydrochloride 25 mg IV, or equivalent.
  - ○ **Ranitidine** 50 mg IV, or equivalent.
- Antiemetic (5-HT$_3$ receptor antagonist):
  - ○ **Ondansetron** 8 mg slow IV infusion, or equivalent, followed by 4 mg q12 h PO on days 4-5

**Plitidepsin:**
90 minutes IV infusion

1.5 mg/d, or
2.0 mg/d, or
2.5 mg/d

IV: intravenously; PO: orally; q12 h, every 12 h; H: Histamine receptor; 5-HT$_3$: serotonin receptor 3

**B**

| Protocol version | Date | Description | Patients |
|---|---|---|---|
| V2.0 | Apr 2020 | Requested premedication 20-30 min before plitidepsin (days 1-3):<br>- Diphenhydramine 25 mg IV<br>- Ranitidine 50 mg IV<br>- Dexamethasone 8 mg PO.<br>Recommended antiemetics as per physician discretion. | 9 |
| V5.0 | Aug 2020 | Added requests (days 1-3) of:<br>- Dexamethasone 8 mg IV (instead of oral)<br>- Ondansetron 8 mg IV | 36 |
| V7.0 | Sep 2020 | Extended ondansetron 4 mg/12 h PO days 4-5 | 9 |

**Figure 5. APLICOV-PC: Protocol Treatment and Pre-medication.**
IV, intravenous; PO, oral; 5-HT$_3$, serotonin (5-hydroxytryptamine) receptor 3.

antiviral effect was induced by plitidepsin against the $\alpha$, $\beta$, $\delta$, $\mu$, and $o$ variants of SARS-CoV-2, which are known to bear several mutations affecting the viral spike protein that facilitates viral entry through its interaction with the human ACE2 receptor (27 Preprint).

Using a drug-resistant mutant host factor, White et al demonstrated that the potent in vitro activity of plitidepsin against SARS-CoV-2 was mediated through eEF1A inhibition (8). In addition, they also showed strong antiviral activity in vivo, characterized by a significant reduction in the viral load in lungs, as well as a clear reduction in alveolar and peribronchial inflammation. In addition, their data also supported a 3-d schedule, as the one used in this clinical study (8).

A phase I trial explored the daily dosing of plitidepsin in cancer patients over 5 consecutive days (13). Dose-limiting toxicities occurred in four of eight patients receiving total cumulative actual doses greater than 13 mg (2.6 mg/day). The recommended dose for subsequent phase 2 studies was defined at 1.2 mg/m$^2$/day (6 mg/m$^2$ total dose). The median actual dose received by these seven

patients was 2.2 mg/day (range 1.9–2.7 mg/day), and the median cumulative dose for their first cycle was 11 mg (range 9.6–13.7 mg) over a 5-d period. In the current study, plitidepsin was given daily for 3 consecutive days, and the maximum daily dose was set at 2.5 mg (7.5 mg in total). This study therefore explored a dose that was 68% of the median recommended dose for cancer patients.

Treatment with plitidepsin was well tolerated, with most AEs being mild and transient in nature. There was no relevant hemotoxicity, and the proportion of laboratory abnormalities was consistent with those expected in COVID-19 patients (28). A central laboratory-based analysis performed on serial ECGs from this study indicated that the administration of 1.5, 2.0, and 2.5 mg plitidepsin as a 90-min IV infusion once daily for 3 consecutive days did not induce cardiac dysfunction, with no alterations seen on left ventricular repolarization, atrioventricular conduction or depolarization.

The death rate due to COVID-19 in our study was 6.7%. A recently published meta-analysis on 33 studies on COVID-19 (totaling 13,398

**Table 1.** Patients' baseline characteristics.

| Parameter | 1.5 mg/day[a] (n = 15) | 2.0 mg/day[a] (n = 15) | 2.5 mg/day[a] (n = 15) |
|---|---|---|---|
| Age (yr)[b] | 51 (32–75) | 49 (34–71) | 53 (31–84) |
| Gender—N (%) | | | |
| Male | 11 (73.3%) | 11 (73.3%) | 8 (53.3%) |
| Female | 4 (26.7%) | 4 (26.7%) | 7 (46.7%) |
| Race—N (%) | | | |
| White | 13 (86.7%) | 9 (60%) | 9 (60%) |
| Latino | 2 (13.3%) | 4 (26.7%) | 6 (40%) |
| Asian | 0 (0%) | 1 (6.7%) | 0 (0%) |
| Arab | 0 (0%) | 1 (6.7%) | 0 (0%) |
| Time from symptom onset to first administration (d)[b] | 6 (3–10) | 6 (3–10) | 6 (2–10) |
| Comorbidities—N (%) | | | |
| One | 2 (13.3%) | 7 (46.7%) | 6 (40%) |
| Two or more | 8 (53.3%) | 6 (40%) | 7 (46.7%) |
| Hypertension | 2 (13.3%) | 2 (13.3%) | 5 (33.3%) |
| Heart disease | 1 (6.7%) | 0 (0%) | 1 (6.7%) |
| COPD[c] | 1 (6.7%) | 1 (6.7%) | 1 (6.7%) |
| Asthma | 2 (13.3%) | 0 (0%) | 3 (20%) |
| Kidney disease | 0 (0%) | 1 (6.7%) | 0 (0%) |
| Diabetes | 1 (6.7%) | 5 (33.3%) | 2 (13.3%) |
| Obesity | 1 (6.7%) | 5 (33.3%) | 4 (26.7%) |
| Patients assessed at room air—N(%) | 9 (60.0) | 7 (46.7) | 6 40%) |
| $SpO_2$ at room air (%)[b] | 95 (92–99) | 95 (91–97) | 96.5 (94–97) |
| $PaO_2/FiO_2$ ratio[b] | 358 (336–408) | 352 (285–396) | 343 (253–481) |
| $PaO_2/FiO_2$ ratio < 300 – N(%) | 0 | 3 (20.0) | 2 (13.3) |
| Disease severity at entry—N (%) (1) | | | |
| Mild COVID-19 | 2 (13.3%) | 3 (20%) | 1 (6.7%) |
| Moderate COVID-19 | 8 (53.3%) | 7 (46.7%) | 8 (53.3%) |
| Severe COVID-19 | 5 (33%) | 5 (33%) | 6 (40%) |
| D-dimer (ng/ml)[b] | 330 (162–1,081) | 463.5 (200–1,270) | 415 (106–962) |
| Ferritin (ng/ml)[b] | 408 (96.8–1,652.8) | 597 (174–1,055.2) | 363 (12.2–1,647) |
| C-reactive protein (mg/l)[b] | 17.7 (1.2–120.4) | 67.2 (2.1–128) | 32.6 (0.3–120) |
| $log_{10}$ copies/ml viral load median (range)[b] | 6.3 (1.5–9.7) | 6.2 (3.8–7) | 5.7 (1.5–10.6) |
| Day 1 six-point ordinal scale—N (%)[d] (2) | | | |
| 2[d] | 6 (40) | 6 (40) | 4 (26.7) |
| 3[d] | 9 (60) | 9 (60) | 11 (73.3) |

[a]Daily for 3 consecutive days.
[b]Median (range).
[c]Chronic Obstructive Pulmonary Disease.
[d]The six-point scale was defined as follows (24): 1, discharged or having reached discharge criteria (defined as "clinical recovery": normalization of pyrexia, respiratory rate <24 breaths per minute, saturation of peripheral oxygen >94% on room air, and relief of cough, all maintained for at least 72 h); 2, hospital admission but not requiring oxygen supplementation; 3, hospital admission for oxygen therapy (but not requiring high-flow or ventilation support); 4, hospital admission for noninvasive ventilation or high-flow oxygen therapy; 5, hospital admission for extracorporeal membrane oxygenation or invasive mechanical ventilation; 6, death.
$PaO_2$, Oxygen partial pressure in arterial blood (imputed from oxygen saturation as described in Supplemental Data 2). $FiO_2$. Oxygen proportion in inspired air.

**Table 2.  Plitidepsin-related adverse events.**

| Parameter | Pre-amendment[a,b] (n = 9) | | | Post-amendment[c] (n = 36) | | |
|---|---|---|---|---|---|---|
| | Grade 1 | Grade 2 | Grade 3 | Grade 1 | Grade 2 | Grade 3 |
| | N (%) | N (%) | N (%) | N (%) | N (%) | N (%) |
| Nausea | 3 (33.3%) | 2 (22.2%) | — | 11 (30.6%) | 3 (8.3%) | — |
| Vomiting | 2 (22.2%) | — | — | 3 (8.3%) | 2 (5.6%) | — |
| Diarrhea | — | — | — | 1 (2.8%) | 1 (2.8%) | 1 (2.8%) |
| Abdominal pain | — | — | — | 2 (5.6%) | — | — |
| Dyspepsia | — | — | — | 2 (5.6%) | — | — |
| Asthenia | — | — | — | 1 (2.8%) | 1 (2.8%) | — |
| Anorexia | — | — | — | 1 (2.8%) | — | — |
| Chest discomfort | - | - | - | 1 (2.8%) | - | - |
| Temperature regulation disorder | — | — | — | 1 (2.8%) | — | — |
| Dysthermia | — | — | — | 1 (2.8%) | — | — |
| Anaphylactic reaction | — | — | 1 (11.1%) | — | — | — |
| Amylase increased[d] | — | — | — | — | 1 (2.8%) | — |
| Lipase increased[e] | — | — | — | — | 1 (2.8%) | — |
| Decreased appetite | — | — | — | — | 1 (2.8%) | — |
| Dizziness | — | — | — | 2 (5.6%) | — | — |
| Dysgeusia | — | — | — | 2 (5.6%) | — | — |

[a]Relevant amendment #9 was implemented in Protocol v5.0 dated 13 August 2020 (Supplemental Data 2): It modified prophylactic medication before plitidepsin infusion to add ondansetron 8 mg IV slow infusion and changed the route of administration of dexamethasone, from oral to IV. The dose of dexamethasone was 8 mg (calculated as 8 mg dexamethasone phosphate, which is equivalent to 6.6 mg dexamethasone base).
[b]25 plitidepsin IV infusions.
[c]108 plitidepsin IV infusions.
[d]Short lasting, 5 min, retro sternal low intensity pain during first day IV infusion: self-resolved plitidepsin infusion completed days 1, 2, and 3.
[e]Same patient, onset day 2 plitidepsin, self-resolved in 48 h.

patients, excluding critical care-only studies), estimated that in hospitalized patients the mortality rate was 11.5% (95% CI: 7.7–16.9%) (29). Published data from large retrospective studies on patients admitted into Spanish hospitals report death rates between 20% and 28% (30, 31, 32, 33, 34). It should be noted, however, that these analyses were performed on data extracted from the first epidemic wave, and APLICOV-PC was run during the second wave, which might account in differences in the availability of health resources, learning curve, and baseline severity of hospitalized patients.

Preliminary efficacy data gathered from this clinical trial were in agreement with the preclinical antiviral activity of plitidepsin described above. Patients treated with plitidepsin showed reductions in viral load with respect to their baseline value, and analytical improvement of biomarkers associated to inflammatory processes (Figs S5 and S6). There were reports of prompt clearance of pneumonia infiltrates in some participants with available chest imaging performed for medical reasons (i.e., not per protocol) (Fig S9A and B). Nevertheless, these results should only be considered suggestive given the limitations of this trial. Studies on the natural history of COVID-19 have shown that viral load peaks at symptom onset, followed by a subsequent decline (35, 36, 37). Therefore, in the absence of a control group, we are not able to conclude that plitidepsin causally affected viral load in this study. In addition, hospital discharge rates, changes in inflammatory biomarkers, and radiological

studies were not predefined end points in this trial. Thus, this study was not designed to evaluate if plitidepsin can improve these parameters and ongoing and future controlled clinical studies will address these hypotheses. The trial did not generate enough information to select the best dose for further clinical development, and further research is currently in progress to address this issue.

The antiviral mechanism of plitidepsin may represent significant advantages in the treatment of COVID-19. Specifically, the likelihood of developing treatment-resistant SARS-CoV-2 strains seems to be remote given that plitidepsin does not directly target a viral component. Therefore, SARS-CoV-2 variants that carry mutations to viral components may be equally sensitive to plitidepsin treatment.

All patients in this study received dexamethasone alongside plitidepsin (i.e., for at least 3 d), which impedes our ability to fully understand the efficacy of plitidepsin in patients with COVID-19. Dexamethasone is currently recommended for hospitalized patients with COVID-19 who have $SpO_2 < 94\%$, including patients on supplemental oxygen, or in patients with critical illness (38). On the other hand, supra-physiologic glucocorticoid treatment leads to lymphocyte apoptosis mediated by the glucocorticoid receptor (39). Noteworthy, there is evidence that the use of corticosteroids is associated with a delayed viral clearance in patients COVID-19 (40). In addition, treatment with glucocorticoids in patients with COVID-19 has been reported to induce an early drop in absolute lymphocyte counts, with

**Table 3.  Summary of protocol-specified efficacy end points.**

| End point | Dose cohort | | | |
| --- | --- | --- | --- | --- |
| | 1.5 mg (N = 14[a]) | 2.0 mg (N = 15) | 2.5 mg (N = 15) | Total (N = 44) |
| Mortality from Day 1 to | | | | |
| Day 7 | — | — | — | — |
| Day 15 | — | — | — | — |
| Day 31[b] | 1 (7.1) | — | 1 (6.7) | 2 (4.5) |
| Patients requiring invasive mechanical ventilation and/or intensive care unit admission | | | | |
| Day 1 to Day 7 | 2 (14.3) | 1 (6.7) | 2 (13.3) | 5 (11.4) |
| Day 8 to Day 15 | 1 (7.1) | 1 (6.7) | 1 (6.7) | 3 (6.8) |
| Day 16 to Day 31 | 1 (7.1) | 1 (6.7) | 1 (6.7) | 3 (6.8) |
| Day 1 to Day 31 | 2 (14.3) | 1 (6.7) | 3 (20.0) | 6 (13.6) |
| Patients requiring noninvasive mechanical ventilation | | | | |
| Day 1 to Day 7 | 4 (28.6) | 0 | 1 (6.7) | 5 (11.4) |
| Day 8 to Day 15 | 3 (21.4) | 0 | 2 (13.3) | 5 (11.4) |
| Day 16 to Day 31 | 1 (7.1) | 1 (6.7) | 1 (6.7) | 3 (6.8) |
| Day 1 to Day 31 | 5 (35.7) | 1 (6.7) | 2 (13.3) | 8 (18.2) |
| Patients requiring oxygen therapy at | | | | |
| Day 7 | 12 (85.7) | 12 (80.0) | 11 (73.3) | 35 (79.5) |
| Day 15 | 4 (28.6) | 1 (6.7) | 4 (26.7) | 9 (20.5) |
| Day 31 | 0 | 2 (13.3) | 1 (6.7) | 3 (6.8) |
| Day 1 to Day 31 | 12 (85.7) | 12 (80.0) | 11 (73.3) | 35 (79.5) |
| Mean change in viral load from baseline to[c] | $log_{10}$ copies/ml | | | |
| Day 4 | −1.23 | −1.49 | −1.32 | −1.35 |
| Day 7 | −2.55 | −2.26 | −2.25 | −2.35 |
| Day 15 | −4.22 | −2.70 | −2.92 | −3.25 |
| Day 31 | −4.70 | −3.53 | −3.49 | −3.85 |
| Mean time from baseline until undetectable viral load[c] | Days | | | |
| | 11 | 14 | 14 | 13 |

[a]One patient who experienced an anaphylactic reaction during the first plitidepsin infusion had treatment discontinued and was not considered evaluable for efficacy.
[b]One additional patient treated at 2.5 mg/day died on Day 57, because of COVID-19 complications.
[c]Results based on 42 patients at Day 4 (13 at 1.5 mg, 14 at 2.0 mg, 15 at 2.5 mg), 40 patients at Day 7 (13 at 1.5 mg, 14 at 2.0 mg, 13 at 2.5 mg), 38 patients at Day 15 (12 at 1.5 mg, 13 at 2.0 mg, 13 at 2.5 mg), and 39 patients at Day 31 (11 at 1.5 mg, 14 at 2.0 mg, 14 at 2.5 mg).

prolonged lymphopenia, along with a significant steady increase in the absolute neutrophil count, altogether driving significant increases in the neutrophil-lymphocyte ratio for at least 2 wk (41). Absolute neutrophil count elevation seen in response to corticosteroid administration is similar to trends associated with increased mortality in several coronavirus studies to include the current SARS-CoV-2 pandemic (41). Notably, in the present study, despite a large proportion of patients receiving dexamethasone for more than 3 d, patients exhibited a median increase in the absolute number of lymphocytes, along with a drop in NLR and in viral RT-PCR results. Again, the lack of a control group prevents conclusive evidence, but this could be a very interesting finding given that, in addition to glucocorticoid therapy, SARS-CoV-2 infection can also induce an early functional exhaustion of cytotoxic lymphocytes that may be responsible for delaying immune responses (42). Future

studies are needed to evaluate the full benefit/risk profile of the concomitant use of glucocorticoids and plitidepsin and eventually define the appropriate candidate for this therapy.

Elevation of inflammation markers such as CRP is associated with an increased risk of disease severity and mortality (43, 44). In this regard, it is noteworthy that in patients with moderate COVID-19 at baseline, intra-patient variations of inflammation markers trended favorably with higher doses of plitidepsin, which may suggest a drug-effect. These observed changes in inflammatory biomarkers may partly explain the rapid clearance of lung infiltrates observed in chest imaging (Fig S9A–C), and is in line with the preclinical observations reported by White et al that treatment with plitidepsin prevents severe lung inflammation (8) (Fig S10A–C).

Translational research on chronic lymphocytic leukemia has identified that plitidepsin can induce cytotoxicity in monocytes at

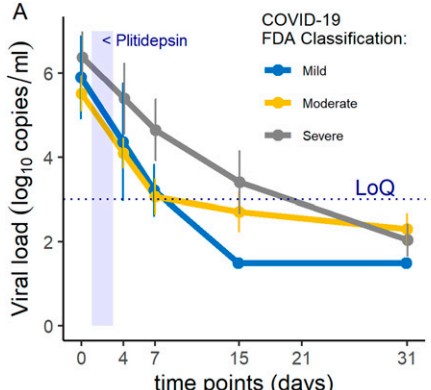

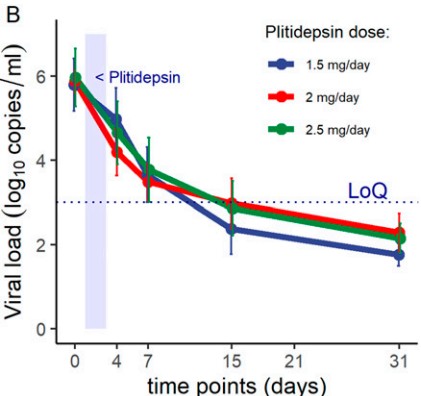

**Figure 6. APLICOV-PC Study: Preliminary Efficacy Outcomes.**
**(A)** Viral load kinetics (qRT-PCR from nasopharyngeal exudates), by baseline severity of the disease (23). **(B)** Viral load kinetics (qRT-PCR from nasopharyngeal exudates), by dose of plitidepsin. LoQ: limit of quantification. See Table S4 for individual data results.

nanomolar concentrations that have little effect in normal lymphocytes (45). Monocytes and macrophages may be infected by SARS-CoV-2, which results in an impairment of the adaptive immune responses against the virus, virus spread, and local tissue inflammation, mediated by the production of large amounts of pro-inflammatory cytokines and chemokines (46). We hypothesize that plitidepsin, besides acting as an antiviral agent, may also modulate immune response by its effects on monocytes/macrophages.

Plitidepsin was hypothesized to provide potential benefits against coronavirus infections, and therefore against SARS-COV-2. This hypothesis was first successfully tested through various in vitro studies, which have led to in vivo tests that have again demonstrated the high potency of plitidepsin in inhibiting SARS-CoV-2. Notably, the putative mechanism of action by which plitidepsin inhibits SARS-CoV-2 (i.e., via a host protein target) suggests that the drug could have a substantial impact on SARS-CoV-2 variants, including the current variants of concern δ and o. The set of in vitro research presented in this work confirms this point.

We now report a proof-of-concept clinical study, showing the safety of administering plitidepsin at the doses and duration described here, and suggesting a potential therapeutic benefit in patients with COVID-19. Nevertheless, our study has several limitations, including the small number of patients evaluated, the large observed variability, and the lack of a control group. These characteristics prevent us from understanding the efficacy of plitidepsin in patients with COVID-19, limit our ability to observe the presence of significant dose–response effects, and restrict our use of this data to hypothesis generation for future studies. An international controlled Phase III trial exploring the efficacy and safety of plitidepsin in hospitalized patients with moderate COVID-19 requiring oxygen supplementation (NEPTUNO; NCT04784559) is currently ongoing. Based on the favorable safety profile seen in this study, doses of 1.5 and 2.5 mg/day plitidepsin have are being evaluated in the Phase III trial.

In summary, we have integrated preclinical and clinical studies on the use of plitidepsin to treat SARS-CoV-2 and other coronavirus infections and generated promising patient data supporting the launch of a Phase III clinical study to demonstrate the efficacy of treatment with plitidepsin in moderate COVID-19 patients who require oxygen therapy.

# Materials and Methods

## Preclinical studies on plitidepsin antiviral activity

### Enjuanes team studies

**Cells and viral infection (HCoV-229E and SARS-CoV)** Monkey Vero E6 cells were kindly provided by E. Snijder (Leiden University Medical Center). Human liver–derived Huh-7 cells were kindly provided by R Bartenschlager (University of Heidelberg). Cells were cultured in DMEM (Lonza) supplemented with 25 mM Hepes, 10% FBS (HyClone), 2% glutamine, and 1% nonessential amino acids (Sigma-Aldrich) and maintained at 37°C in a humidified atmosphere of 5% $CO_2$.

HCoV-229E was kindly provided by V Thiel (Institute of Virology and Immunology). SARS-CoV virus was rescued from the corresponding infectious cDNAs (47). All experiments with SARS-CoV infectious virus were performed in BSL-3 facilities at CNB-CSIC according to institutional guidelines.

**Experimental design** Human Huh-7 cells were infected with HCoV-229E-GFP virus at an MOI of 0.01. After 8 hours post infection (hpi), the medium was replaced by fresh medium containing different plitidepsin concentrations. The presence of fluorescent foci, indicating HCoV-229E-GFP infection, was analyzed at 48 hpi.

SARS-CoV virus stock at $2 \times 10^7$ pfu per ml was used. Confluent Vero E6 cells were infected with SARS-CoV at a MOI of 0.01. After 1 hpi, virus inoculum was retired and medium was replaced by fresh DMEM-HEPES medium with 2% FBS at different concentrations of the compounds. Treatments were as follows: (1) mock-infected cells and SARS-CoV–infected cells in the absence of DMSO were used as a control to rule out DMSO toxicity effect; (2) DMSO at the same % present in the compound dilutions was used as a negative control; (3) plitidepsin and didemnin B (data not shown), the most promising compounds in the previous HCoV-229E-GFP screening, were tested at 0.5, 5, 50, 100, and 500 nM; (3) PM021473 was used as a control with no effect on CoV infection. It was tested at 0.5, 5, 50, 100, and 500 (data not shown); (4) remdesivir was used as a positive control, and tested at 0.5, 5, 100, 500, and 1,000 nM.

Two independent biological replicates were performed for each condition. At 48 hpi, culture supernatant was collected for virus titration, following standard procedures (48).

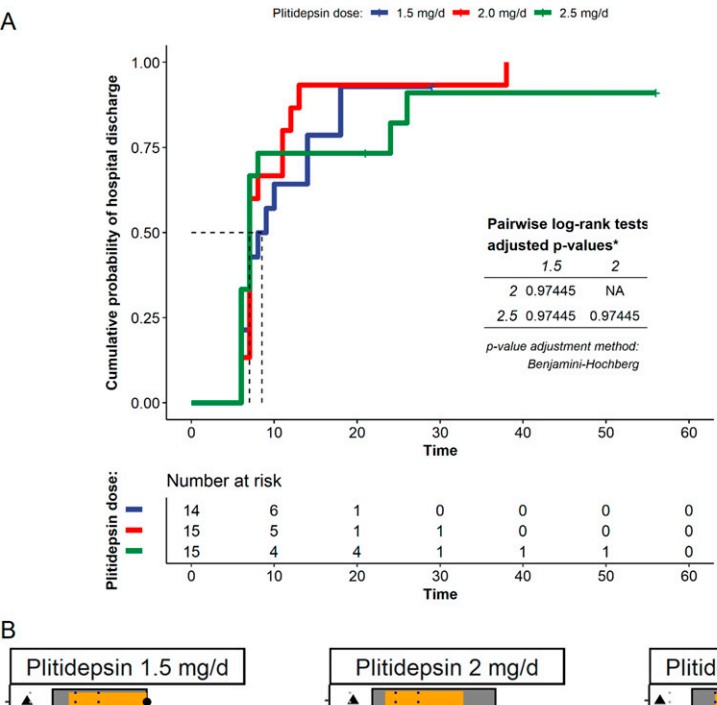

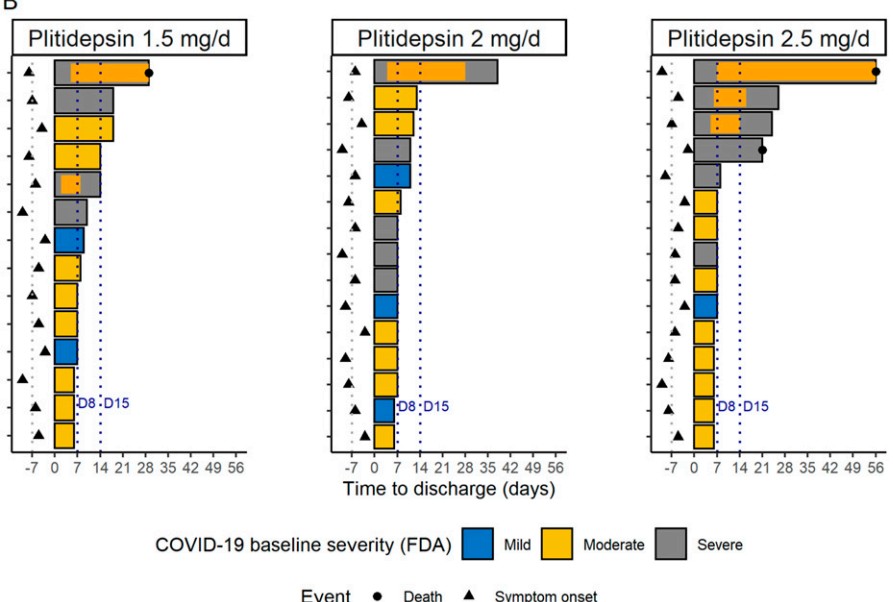

**Figure 7. Post hoc analysis on hospital discharge by plitidepsin dose.**
**(A)** Reverse Kaplan Meier plot showing the cumulative incidence of hospital discharge by plitidepsin dose. **(B)** Length of hospitalization, by plitidepsin dose and disease severity at baseline (23). Orange bars represent admission in intensive care units. Dashed lines labeled D8 and D15 are days 8 and 15, respectively, considering the start of therapy with plitidepsin as Day 1; this is equivalent to stays of 7 or 14 d from the start of therapy. See Figs S2S–SFigs S2–S4 for post hoc analysis on hospital discharge and respiratory support according to the severity of the disease at baseline.

Cells were also collected and total intracellular RNA was purified using RNeasy Mini Kit (QIAGEN). Total cDNA was synthesized using 100 ng of total RNA as a template, random hexamers, and a high-capacity cDNA transcription kit (Life Technologies). SARS-CoV genomic RNA was evaluated using a custom TaqMan assay targeting nsp2 sequence. The human hydroxymethylbilane synthase (HMBS) gene (TaqMan code Hs00609297_m1) was used as a reference housekeeping gene. Data were acquired with a 7500 real-time PCR system (Applied Biosystems) and analyzed with 7500 software v2.0.6. Relative quantifications were performed using the $2^{-\Delta\Delta Ct}$ method (49).

### Rodon team studies
**Cells and viral isolation** Vero E6 cells (ATCC CRL-1586) were cultured in DMEM (Lonza), supplemented with 5% FCS (EuroClone),

100 U/ml penicillin, 100 μg/ml streptomycin, and 2 mM glutamine (all Thermo Fisher Scientific).

SARS-CoV-2 was isolated from a nasopharyngeal swab and its genomic sequence deposited at GISAID repository (http://gisaid.org) with accession ID EPI_ISL_510689 as previously described (26 Preprint).

**Antiviral activity** Plitidepsin was assayed from 5 to 100,000 nM in duplicates as previously detailed (26 Preprint).

Drug dilutions were added to Vero E6 cells and, immediately after, 20 tissue culture infectious dose 50% (TCID50) per well of SARS-CoV-2 were inoculated to 30,000 cells in 200 μl. This viral concentration achieves a 50% of cytopathic effect 3 d postinfection.

Untreated noninfected cells and untreated virus-infected cells were used as negative and positive controls of infection,

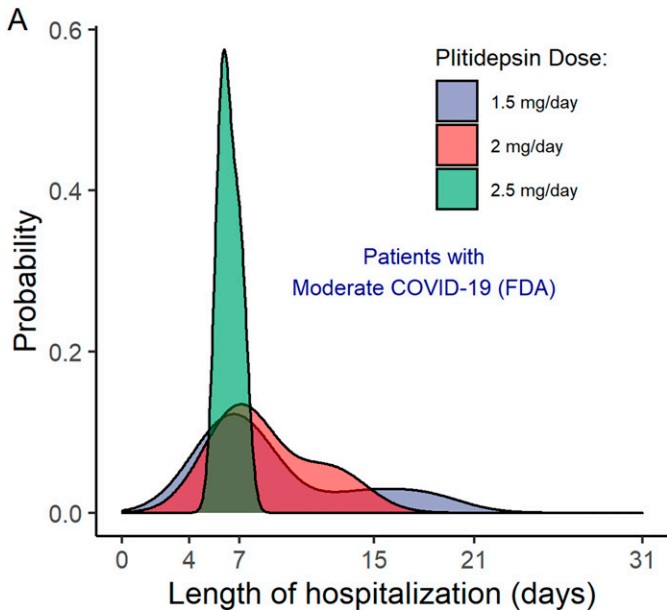

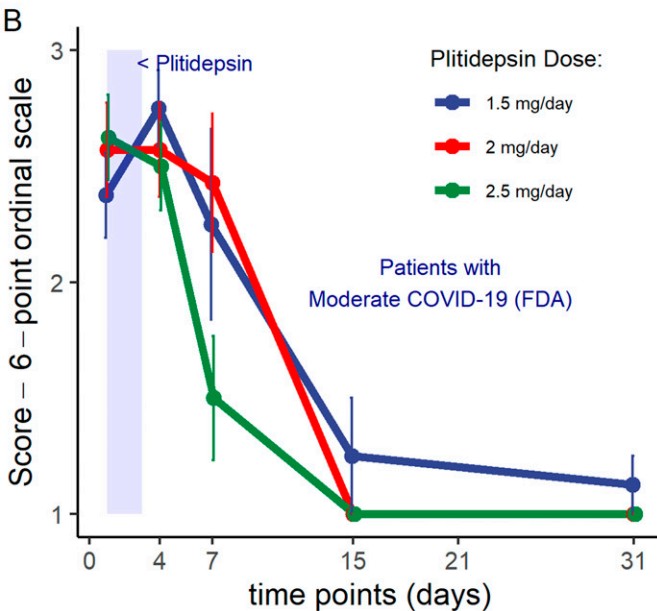

**Figure 8. Post hoc analysis on hospital discharge by plitidepsin dose in patients with moderate COVID-19 at baseline.**
**(A)** Subgroup of patients with moderate COVID-19 (n = 23 pts): Distribution of the probability of the duration of the hospitalization, according to the dose of plitidepsin administered. **(B)** Subgroup of patients with moderate COVID-19 (n = 23 pts): Mean score over time of a six-category ordinal scale in patients with moderate disease at baseline, according to the administered dose of plitidepsin. The six-point scale was defined as follows (24): 1, discharged or having reached discharge criteria (defined as "clinical recovery": normalization of pyrexia, respiratory rate <24 breaths per minute, saturation of peripheral oxygen >94% room air, and relief of cough, all maintained for at least 72 h); 2, hospital admission but not requiring oxygen supplementation; 3, hospital admission for oxygen therapy (but not requiring high-flow or ventilation support); 4, hospital admission for noninvasive ventilation or high-flow oxygen therapy; 5, hospital admission for extracorporeal membrane oxygenation or invasive mechanical ventilation; 6, death. See also Table 3 and Figs S7 and S8.

respectively. To detect any drug-associated cytotoxic effect, Vero E6 cells were equally cultured in the presence of increasing drug concentrations, but in the absence of virus.

Viral-induced cytopathic or drug-induced cytotoxic effects were measured 3 d post infection, using the CellTiter-Glo luminescent cell viability assay (Promega). Luminescence was measured in a Fluoroskan Ascent FL luminometer (Thermo Fisher Scientific).

Cells not exposed to the virus were used as negative controls of infection and were set as 100% of viability to normalize data and calculate the percentage of cytopathic effect. Response curves of compounds were adjusted to a nonlinear fit regression model, calculated with a four-parameter logistic curve with variable slope.

### Krogan/Garcia-Sastre teams' studies

**Cell culture and drugs** Calu-3 cells (ATCC HTB-55) and Caco-2 cells were a kind gift Dr Dalan Bailey (Pirbright Institute). Cells were cultured in DMEM supplemented with 10% heat-inactivated FBS (Labtech), 100 U/ml penicillin/streptomycin, with the addition of 1% Sodium Pyruvate (Gibco) and 1% Glutamax for Calu-3 and Caco-2 cells. All cells were passaged at 80% confluence. For infections, adherent cells were trypsinized, washed once in fresh medium, and passed through a 70 $\mu$m cell strainer before seeding at $0.2 \times 10^6$ cells/ml into tissue-culture plates. Calu-3 cells were grown to 60–80% confluence before infection as described previously (50 Preprint). Plitidepsin (PharmaMar) and remdesivir (SelleckChem) were reconstituted in sterile DMSO.

**Viruses** SARS-CoV-2 strain BetaCoV/Australia/VIC01/2020 (NIBSC) and SARS-CoV-2 B.1.1.7 (SARS CoV 2 England/ATACCC 174/2020) strain were propagated by infecting Caco-2 cells at a MOI 0.01 TCID50/cell, in DMEM supplemented with 10% FBS at 37°C.

Virus was harvested at 72 hpi and clarified by centrifugation at 2,300g for 15 min at 4°C to remove any cellular debris. Virus stocks were aliquoted and stored at –80°C. Virus titers were determined by quantification of SARS-CoV-2 RNA genomes/ml as previously described (50 Preprint).

**Infection and drug assays** Calu-3 and Caco-2 cells were pretreated with remdesivir or plitidepsin at the indicated concentrations or DMSO control at an equivalent dilution for 2 h before SARS-CoV-2 infection.

Caco-2 and Calu-3 cells were infected at $1 \times 10^3$ copies per cell, equivalent to an MOI of 0.01 TCID50 per cell (as titered on Vero.E6). Inhibitors were maintained throughout infection.

Cells were harvested after 24 h for analysis and viral infection measured by intracellular detection of SARS-CoV-2 nucleoprotein by flow cytometry. Tetrazolium salt (MTT) assay was performed to verify cell viability. 10% vol/vol MTT was added to the cell media and cells were incubated for 24 h at 37°C. Cells were lysed with 10% SDS and 0.01M HCl and the formation of purple formazan was measured at 620 nm.

**Flow cytometry** For flow cytometry analysis, adherent cells were recovered by trypsinization and washed in PBS with 2 mM EDTA

(PBS/EDTA). Cells were stained with fixable Zombie NIR Live/Dead dye (BioLegend) for 6 min at room temperature. Excess stain was quenched with FBS-complemented DMEM. Cells were fixed in 4% paraformaldehyde before intracellular staining.

For intracellular detection of SARS-CoV-2 nucleoprotein, cells were permeabilized for 15 min with Intracellular Staining Perm Wash Buffer (BioLegend). Cells were then incubated with 1 μg/ml CR3009 SARS-CoV-2 cross-reactive antibody (a kind gift from Dr. Laura McCoy) in permeabilization buffer for 30 min at room temperature, washed once and incubated with secondary Alexa Fluor 488–Donkey-anti-Human IgG (Jackson Labs). All samples were acquired and analyzed on a NovoCyte 3005 Flow Cytometer System (Agilent).

**In vitro study of SARS-CoV-2 variants sensitivity to plitidepsin** Nasopharyngeal swab specimens were collected as part of the routine SARS-CoV-2 surveillance conducted by Viviana Simon and the Mount Sinai Pathogen Surveillance program (IRB approved, HS#13-00981). Specimens were selected for viral culture on Vero-E6 cells based on the complete viral genome sequence information (51). The SARS-CoV-2 virus USA-WA1/2020 was obtained from BEI resources (NR-52281) and used as wild-type reference. Viruses were grown in Vero-TMPRSS2 cells (BPS Bioscience) for 4–6 d; the supernatant was clarified by centrifugation at 4,000*g* for 5 min and aliquots were frozen at –80°C for long term use. Expanded viral stocks were sequence-verified to be the identified SARS-CoV-2 variant and tittered on Vero-TMPRSS2 cells before use in antiviral assays.

Two thousand (2,000) HeLa-ACE2 cells were seeded into 96-well plates and incubated for 24 h. 2 h before infection, the medium was replaced with a new media containing the compound of interest, including a DMSO control. Plates were then transferred into the biosafety level 3 (BSL-3) facility and 1,000 PFU (MOI = 0.25) of SARS-CoV-2 was added, bringing the final compound concentration to those indicated. SARS-CoV-2/WA1, α (B.1.1.7), β (B.1.351), δ (B.1.617.2), μ (B.1.621), and *o* (B.1.1.529) variants were used as indicated. Plates were then incubated for 48 h. Infectivity was measured by the accumulation of viral NP protein in the nucleus of the HeLa-ACE2 cells (fluorescence accumulation). Percent infection was quantified as ((Infected cells/Total cells) – Background) × 100, and the DMSO control was then set to 100% infection for analysis. Cytotoxicity was also performed at matched concentrations using the MTT assay (Roche), according to the manufacturer's instructions. Cytotoxicity was performed in uninfected HeLa-ACE2 cells with same compound dilutions and concurrent with viral replication assay. All assays were performed in biologically independent triplicates.

**Animal models of SARS-CoV-2 infection experiments and lung histological analysis** These studies were conducted under protocols approved by the Institutional Animal Care and Use Committee (IACUC) and were performed in animal biosafety level 3 (BSL3) facility at the Icahn school of Medicine in Mount Sinai Hospital, New York City. They are described in reference 8.

**Model-based dose justification**

**Definition of in vitro target dose of plitidepsin (Boryung Pharmaceutical (52))**
**Virus and cells** Both Vero (ATCC CCL-81) and Calu-3 cells (ATCC HTB-55) were purchased from the American Type Culture Collection

(ATCC). SARS-CoV-2 (βCoV/KOR/KCDC03/2020) was provided by Korea Centers for Disease Control and Prevention (KCDC) and was propagated in Vero cells. Viral titers were determined by plaque assays in Vero cells.

**Reagents** Plitidepsin (batch#: 16 D19) and ampoule (batch#: 60108) were provided by Boryung Pharmaceutical. Anti-SARS-CoV-2 N protein antibody was purchased from Sino Biological lnc. Alexa Fluor 488–goat anti-rabbit [IgG (H+L) secondary antibody and Hoechst 33342 were purchased from Molecular Probes.

**Drug treatment, infection and immunofluorescence staining** 2 mg of plitidepsin was dissolved in either 1,801.3 μl of DMSO or in ampoule at a final concentration of 1 mM and a twofold dilution series was made with 20-points. Vero cells were seeded at l.2 × 10⁴ cells per well in DMEM supplemented with 2% FBS and I X Antibiotic-Antimycotic solution (Gibco) in black, 384-well, μClear plates (Greiner Bio-One), 24 h before plitidepsin treatment and virus infection. Calu-3 cells were seeded at 2.0 × 10⁴ cells per well in Eagle's Minimel Essential Medium supplemented with 10% FBS and 1X Antibiotic-Antimycotic solution (Gibco) in black, 384-well, μClear plates (Greine r Bio-One), 24 h before plitidepsin treatment and virus infection. Twenty-point plitidepsin dilution series generated above was added to Vero or Calu-3 cells with the highest concentration at 5 μM. After 1 h, the plates were transferred into the BSL-3 containment facility for virus infection. SARS-CoV-2 was added at a MO1 of 0.0125 and 0.1 to the plates for Vero cells and Calu-3 cells, respectively.

The cells were fixed at 24 hpi with 4% paraformaldehyde. Anti-SARS-CoV-2 N protein antibody and Alexa Fluor 488–conjugated goat anti-rabbit IgG antibody were used for immunostaining of viral N protein and Hoechst 33342 were used to stain nuclei of the host cells.

**Image analysis** The images acquired with Operetta (Perkin Elmer) were analyzed using our in-house Image-Mining (IM) software to quantify cell numbers and infection ratio by counting Hoechst-stained nuclei and viral N protein-expressing cells, respectively. The infection ratio of each well was normalized to the average of infection percentage of infection group (0.5% DMSO) and mock infection group in each plate. The cell ratio was determined according to the number of cells of each well versus the average number of cells of mock infection in each plate and described as "cell number to mock" in the dose–response curve (DRC) graph. The DRCs of plitidepsin (both. dissolved either in DMSO or in ampoule), was analyzed using the XLfit equation: Y = Bottom + (Top – Bottom)/ $(1 + (IC_{50}/X)Hillslope)$.

The $IC_{50}$ and 50% toxicity concentration (CCso) values were calculated from the fitted DRCs. Selectivity index (SI) was calculated as $CC_{50}/IC_{50}$. All $IC_{50}$ and $CC_{50}$ values were determined in duplicate experiments. All the experiments were conducted simultaneously.

*Plasma protein binding study*
The binding of plitidepsin to plasma proteins was determined in vitro at three concentrations (100, 250 and 500 ng/ml) in male rat, dog and human plasma (53).

Rat (CD), beagle dog, and human blood was collected at Aptuit Srl from at least three male donors, and plasma prepared by

centrifuging blood at 2,000*g*, 4°C for 10 min. Human plasma was obtained from healthy and fasted male volunteers. Na-heparin was used as the anti-coagulant for all species. All plasma was stored at approximately –20°C and thawed only once, on the day of the experiment.

Plasma protein binding was assessed using Rapid Equilibrium Dialysis plates pre-loaded with equilibrium dialysis membrane inserts (MWCO ~8 kD).

Separation of free compound from protein-bound material was achieved by dialysis of the sample through the membrane under appropriate shaking at 37°C. The suitability of equilibrium dialysis as a method for protein binding determination of plitidepsin was assessed.

Before experimentation, the time required to reach equilibrium was determined in male rat and human plasma at one concentration (250 ng/ml) and six selected time points (2, 3, 4, 5, 6, and 7 h) at 37°C.

Plitidepsin stability was assessed in plasma (all species) and PBS at two concentrations (100 and 500 ng/ml for plasma; 0.5 and 50 ng/ml for PBS) by comparing plitidepsin concentration in spiked and preincubated plasma before and after incubation at 37°C for 5 h as determined previously.

### Plitidepsin tissue distribution study

An in vivo distribution study was carried out in rats. PM140064 (14C1-Plitidepsin) was supplied by PharmaMar at a radiochemical purity of 95% or greater, with no single impurity 3% or greater. The study was conducted at Aptuit Srl (54).

24 male and 24 female Sprague–Dawley rats each received a single IV bolus administration of [14C] Plitidepsin at a target dose of 0.2 mg/kg. After administration, three animals per time-point (0.25, 1, 2, 4, 8, 24, 48, and 72 h post-dose) were exsanguinated from the abdominal aorta and blood retained. Actual times of bleeding were recorded. In addition to blood, the following tissues were collected: brain, eyeballs, heart, liver, lung, skeletal muscle (quadriceps), fat, kidneys, stomach, skin, small intestine, spleen, thyroid, lymph nodes, and testes or ovaries. At the end of the collection period animals were euthanized by $CO_2$ asphyxiation and the carcasses were discarded. Additional animals were bled to obtain control blood/plasma and tissues.

### Estimation of target $IC_{50}$ and $IC_{90}$ total plasma concentrations

The results from Boryung Pharmaceuticals in infected Vero cells demonstrated a very strong antiviral effect induced by plitidepsin, with half-maximal inhibitory concentration ($IC_{50}$) as low as 3.26 (95%, 2.97–3.59) nM. These were consistent with results from a different cell line, Calu-3. Based on the resulting Hill slope (2.08) and according to the following equation (55), a 90% maximal inhibitory concentration ($IC_{90}$) of 9.38 (95%CI, 7.65–11.50) nM was estimated.

$$ICF_{invitro} = \left( \frac{F}{100 - F} \right)^{1/H} \times IC50_{invitro}$$

where F is the percentage of response (i.e., 90%) and H is the Hill slope.

Results from previously described studies show that: (a) in human plasma, plasma-protein binding of plitidepsin was estimated at 98%, independent of drug concentration (Table S2), and (b) after the administration of a single i.v. bolus dose (0.2 mg/kg) of plitidepsin (14C-labeled), a significant increased distribution of radioactivity was found in lung; this preferential distribution led to lung-to-plasma area under the curve ratio (LPR) of ~543-fold, respectively (calculated from Table S3).

The partition coefficient LPR enables the quantification of the total drug concentration in the tissue, and, by assuming similar lung distribution in humans to that observed in rodents and similar fraction of unbound drug in plasma and tissue, unbound exposures in human lung can be further estimated.

Therefore, total plasma concentration ($\mu g/l$) of plitidepsin associated with lung exposures above the in vitro target concentration $IC_{50}$ and $IC_{90}$, were estimated at 0.33 (95% CI, 0.30–0.37) $\mu g/l$ and 0.96 (95% CI, 0.78–1.18) $\mu g/l$, respectively, according to the following equation:

$$ICF_{total,plasma} = \frac{ICF_{total,in\ vitro}}{f_{u,human} \cdot LPR_{rat}},$$

where $ICF_{total,plasma}$ is the total target plasma concentration ($\mu g/l$), $ICF_{total,in\ vitro}$ is the concentration ($\mu g/l$) used in the in vitro experiment, $f_{u,human}$ is the unbound fraction in human plasma, and $LPR_{rat}$ is the lung-plasma area under the curve (AUC) ratio in the distribution study in rats. This equation considers plasma protein binding and lung-plasma AUC ratio, based on current recommendations (21).

### Simulation of plitidepsin plasma exposures at the selected dose regimen

A validated pharmacokinetic population model of plitidepsin (22) updated with data from multiple myeloma patients, was used to simulate plitidepsin plasma profiles in typical subjects treated with the selected dose regimen (days 1–3 in 1 h 30 min infusion), to observe whether they would reach the estimated target plasma concentrations with antiviral activity. This model was developed based on plasma and blood concentrations from 549 cancer patients from four Phase I, nine Phase II and one Phase III study treated with plitidepsin either as monotherapy or in combination with dexamethasone, at doses ranging from 0.8 to 8.0 mg/m² as a 1- or 24-h infusion weekly, 3- or 24-h infusion biweekly, or 1-h infusion daily for 5 consecutive days every 3 wk. An open, three-compartment disposition model with linear elimination and linear distribution from the central compartment to two peripheral compartments was used to describe the pharmacokinetics (PK) of plitidepsin in plasma. The model was parameterized in terms of systemic clearance (Cl), central (V1) and peripheral volume of distribution for the shallow (V2) and deep (V3) compartments, and intercompartmental exchange flows for shallow (Q2) and deep (Q3) compartments. The concentration of plitidepsin bound to red blood cells was modeled as a nonlinear function and the plitidepsin blood concentration was estimated according to the following equation (22):

$$C_{blood} = C_{plasma} \cdot (1 - HCT) + \frac{B_{max} \cdot C_{plasma}}{k_d + C_{plasma}} \cdot HCT$$

where $B_{max}$ corresponds to the maximal plitidepsin concentration bound to blood cells, $k_d$ is the plitidepsin plasma concentration at which the plitidepsin bound to red blood cells is half-maximal, and HCT is the baseline hematocrit of each patient.

Based on the deterministic simulations of plasma concentration–time profiles, flat doses of 1.5, 2.0, and 2.5 mg will be associated to plasma concentrations above $IC_{50}$ throughout the whole treatment period and will remain above $IC_{90}$ during most of the administration interval, as depicted in Fig 3, whereas accumulation after three repeated administrations is minimal.

## Proof-of-concept study design

### Trial design

The plitidepsin clinical proof-of-concept study (APLICOV-PC, APL-D-002-20; EudraCT #2020-001993-31; NCT04382066) was a multicenter, randomized, parallel, open-label, proof-of-concept clinical trial, exploring three dose levels of plitidepsin (1.5, 2.0, and 2.5 mg/day, flat doses) for 3 consecutive days, as a 90-min i.v. infusion, in adult patients with COVID-19 who required hospitalization. The study was conducted in 10 hospital centers in Spain between 12 May 2020 and 26 November 2020. Presented data reflect all analyses completed by the cut-off date of 10 December 2020. The research was performed in accordance with good clinical practice guidelines and adhered to the tenets of the Declaration of Helsinki. The study protocol is available in Supplemental Data 1. Supplemental Data 2 summarizes the protocol amendments as well as post hoc analyses that were performed.

### Participants

Adult (aged ≥18 yr) hospitalized patients with a positive SARS-CoV-2 infection confirmed by real-time RT-PCR testing of a nasopharyngeal exudate or the lower respiratory tract were eligible for inclusion in this study. Despite the eligibility of patients with samples from the lower respiratory tract, all patients reported here had only nasopharyngeal swab samples. Included patients must have had the onset of SARS-CoV-2 infection symptoms no more than 10 d before initiating the study. Women of reproductive age had a negative pregnancy test and were non-lactating. Women and men with partners of childbearing potential agreed to take effective contraception while on study and for 6 mo after the last dose of plitidepsin. All included patients provided written informed consent before the initiation of the study.

Key exclusion criteria included participation in another COVID-19–related clinical trial; receiving antivirals, chloroquine or derivatives, IL-6 receptor inhibitors, or immunomodulatory drugs for treatment of COVID-19; or requiring mechanical ventilation support at enrollment. Patients were also excluded for evidence of multi-organ failure, clinically relevant heart disease, clinically relevant arrhythmia or history of QTc interval prolongation, or neuropathy ≥ grade 2. Full exclusion criteria can be found in Supplemental Data 1.

### Randomization

Randomization was done through registration in the study's electronic case report form. The system assigned a unique randomization number for each patient. Patients were enrolled sequentially and allocated by block randomization into the three dose groups (1.5, 2.0, and 2.5 mg plitidepsin) up to an initial sample size of nine patients per group. To ensure safety, the first three patients in the 2.0 mg group could not be enrolled until the first three patients in the 1.5 mg cohort had successfully completed Day 15 assessments (12 d after completing plitidepsin dosing); similarly, the first three patients in the 2.5 mg cohort could be not enrolled until the first three patients in the 2.0 mg cohort had successfully completed Day 15 assessments (12 d after completing plitidepsin dosing). Finally, three patients had to be treated in the 2.5 mg cohort to complete the accrual of nine patients per cohort. When multiple dose levels were open for enrollment, patients were randomized 1:1 or 1:1:1 as appropriate (Fig 4). After an interim safety analysis, a protocol amendment was approved to subsequently expand the accrual up to 45 patients (six additional patients per dose group).

### Interventions

Enrolled patients received plitidepsin administered as an i.v. infusion over 90 min on days 1, 2, and 3 at dose levels of 1.5, 2.0, and 2.5 mg. A single batch (Lot 16D19) was administered to all patients. All patients also received, 20–30 min before the infusion of plitidepsin (i.e., on days 1–3), a premedication regimen consisting in: diphenhydramine 25 mg i.v., ranitidine 50 mg i.v., and dexamethasone 8 mg orally. In a protocol amendment dated August 2020, the prophylactic medication was modified, replacing oral dexamethasone with i.v. administration of dexamethasone 8 mg (calculated as 8 mg dexamethasone phosphate, which is equivalent to 6.6 mg dexamethasone base), and adding ondansetron 8 mg. After a subsequent protocol amendment (September 2020), ondansetron 4 mg oral was administered every 12 h until 48 h after the last administration of plitidepsin (Fig 5 and Supplemental Data 2).

Concomitant administration of IL-6 receptor inhibitors, immunomodulatory drugs, and chloroquine and derivatives were not permitted. Other pharmacological treatments for COVID-19 were allowed, at the discretion of the investigator, after 24 h from the administration of the last dose of plitidepsin. Potent CYP3A4 inhibitors and inducers had to be discontinued before starting treatment with plitidepsin and could be re-administered 24 h after its last dose.

### Outcomes

The primary outcome was safety, which included the proportion of patients who developed AEs as graded in terms of severity (NCI CTCAE v.5.0). Changes from baseline in vital signs; complete blood count; coagulation parameters; serum chemistry (alanine aminotransferase [ALT], aspartate aminotransferase [AST], γ-glutamyl transferase [GGT], total bilirubin, alkaline phosphatase, lactate dehydrogenase [LDH], creatinine phosphokinase [CPK], ferritin, troponin, and C-reactive protein [CRP]); and 12-lead ECGs for PR and QT intervals were also measured. With the exception of vital signs, which were measured every 8 h during hospitalization then at protocol-specified visits through Day 31, other safety evaluations occurred at baseline, days 1–7, 15, and 31.

Secondary outcomes included change from baseline of viral load, as evaluated by quantitative PCR via central laboratory on nasopharyngeal exudate or sampled from the lower respiratory tract within 48 h before first administration of plitidepsin and on days 4, 7, 15, and 31, and the percentage of patients requiring oxygen therapy, noninvasive mechanical ventilation, invasive mechanical ventilation and/or intensive care unit admission, and the percentage of patients with a fatal outcome on days 7, 15, and 31.

### SARS-CoV-2 quantification

Centralized assessments of nasopharyngeal samples were performed at SYNLAB. Viral RNA extraction from the samples was performed using the Maxwell HT Viral TNA (AX2340) Promega, on the Hamilton automated extraction platform. Detection of nCoV2019 was carried out with the kit: TaqPath TM COVID-19 RT-PCR KIT (A48102) Thermo Fisher Scientific. The positivity/negativity criteria used for the interpretation of the results are defined in Table S7.

Each sample was analyzed in triplicate, and the cycle threshold (CT) value obtained was extrapolated to the standard curve, so that a value of viral load is obtained per reaction for said sample. The quantification, in triplicate, of the viral load of the patient sample was carried out at each of the five time points.

For quantification, in each sample plate, eight points, in triplicate (24 PCR in total), of the plasmid IDT 2019 nCov Kit CDC EUA of known viral load was used to generate the standard line on which to extrapolate the results of the samples of the trial patients. The sensitivity of the analytical method is 10 mRNA copies/run. Assuming a starting sample volume of 200 $\mu$l, from which 100 $\mu$l of mRNA can be obtained, and that a single PCR run requires 5 $\mu$l mRNA, it can be estimated that the sensitivity for 100% of the replicates is 1,000 mRNA copies/ml (3 $\log_{10}$). For mathematical analysis and graphical visualization, an intermediate value of 1.5 $\log_{10}$ has been imputed to samples with undetected SARS-CoV-2 mRNA.

### Statistical methods

Analysis of safety data was conducted on all patients who received at least one dose of study drug and the results were reported using descriptive statistics by means of frequency and percentage of patients for categorical variables and medians (range and interquartile range [IQR])/mean (SD) for continuous variables.

AEs were coded by using Medical Dictionary for Regulatory Activities (version 17.0) and were presented as grouped by system organ class and preferred term. The associated grades with each of the AEs were evaluated using the National Cancer Institute-Common Terminology Criteria for AEs (version 5.0) toxicity criteria.

Analysis for preliminary evidence of efficacy was performed on all patients who had efficacy assessments at baseline and at least one subsequent assessment during treatment.

Post hoc analyses were conducted to (1) assess disease severity at baseline (mild, moderate, severe COVID-19 by FDA criteria) (23) and (2) assess covariates associated with efficacy end points, including hospital discharge by Day 15 or Day 8.

The Kaplan–Meier method was used for the description of time to event end points and the median and the 95% confidence intervals were provided.

In regards with subgroup analyses, safety and efficacy outcomes also were assessed by age-group (<65 versus ≥65 yr) and efficacy outcomes were assessed by disease severity at baseline. Safety outcomes were also assessed before and after the implementation of the protocol amendment.

Continuous and categorical end points for the treatment effect are reported. The differences in proportions are reported with 95% confidence intervals. Formal statistical testing between subgroups were not predefined because of the reduced numbers in each subgroup. Exploratory stepwise logistic regression and Cox regression models have been built to select the independent prognostic factors of higher statistical significance to explain the more relevant efficacy outcomes using a threshold of $\alpha$ 0.10 for variable selection. All statistical analyses and plots were carried out using SAS software 9.4 (SAS Institute) and R 4.0.3.

## Data Availability

Plitidepsin is available from PharmaMar for noncommercial use under a materials transfer agreement. All relevant data are included within this manuscript and all materials other than plitidepsin are readily available upon request from the corresponding authors. This work is licensed under a Creative Commons Attribution 4.0 International (CC BY 4.0) license, which permits unrestricted use, distribution, and reproduction in any medium, provided the original work is properly cited. To view a copy of this license, visit https://creativecommons.org/licenses/by/4.0/. This license does not apply to figures/photos/artwork or other content included in the article that is credited to a third party; obtain authorization from the rights holder before using such material.

## Supplementary Information

## Acknowledgements

We are indebted to the women and men that gave their consent to participate in this study, and to their relatives, for understanding their decision in these exceptional circumstances. We would like to thank Pascal Besman for his input, Timothy Silverstein for providing editorial support, and Lorena Martin Peña for technical and secretarial assistance. The full clinical research teams at each one of the participating sites have played an instrumental role. We recognize hard work and commitment done by Paz Cañadas, Daniel Sánchez-Brualla, and Carlota Costa from Synlab Diagnósticos Globales, SAU as central laboratory in charge of viral RT-PCR assessments. We appreciate the collaboration with Boryung Pharmaceuticals in the first in vitro study on the activity of plitidepsin in SARS-CoV-2 infection models. We thank R Albrecht for support with the BSL-3 facility and procedures at the Icahn School of Medicine at Mount Sinai, New York. We thank Ana Tercero, Angelines Barroso, Barbara Garcia, Cora Loste, Daniel Perez-Zsolt, Ester Ballana, Gemma Lladós, Hervé Dhellot, Ivette Casafont, Joaquim Segalés, Jon Cendoya, Jordana Muñoz-Basagoiti, José Ramón Santos, Laura Soldevila, Lola Castro, Lourdes Mateu, Lucía Gutiérrez-Chamorro, María Luisa Ramírez, Miriam Ramírez, Rafael Fernández Alonso, and Sonia Extremera for

their dedication and commitment. Funding: This study has been funded by Pharmamar, SA (Madrid, Spain). This work was supported by grants from the Government of Spain (PIE_INTRAMURAL_ LINEA 1 - 202020E079; PIE_INTRAMURAL_CSIC-202020E043). The research of CBIG consortium (constituted by IRTA-CReSA, BSC, & IrsiCaixa) is supported by Grifols pharmaceutical. We also acknowledge the crowdfunding initiative #Yomecorono (https://www.yomecorono.com). N Izquierdo-Useros has nonrestrictive funding from PharmaMar to study the antiviral effect of Plitidepsin. NJ Krogan was funded by grants from the National Institutes of Health (P50AI150476, U19AI135990, U19AI135972, R01AI143292, R01AI120694, and P01AI063302); by the Excellence in Research Award (ERA) from the Laboratory for Genomics Research (LGR), a collaboration between the University of California, San Francisco (UCSF), University of California, Berkley (UCB), and GlaxoSmithKline (GSK) (#133122P); by the Roddenberry Foundation, and gifts from QCRG philanthropic donors. This work was supported by the Defense Advanced Research Projects Agency (DARPA) under Cooperative Agreement #HR0011-19-2-0020. The views, opinions, and/or findings contained in this material are those of the authors and should not be interpreted as representing the official views or policies of the Department of Defense or the U.S. Government. This research was partly funded by Center for Research for Influenza Pathogenesis and Transmission (CRIPT), a National Institute of Allergy and Infectious Diseases (NIAID) supported Center of Excellence for Influenza Research and Response (CEIRS, contract # 75N93021C00014), by DARPA grant HR0011-19-2-0020, by supplements to NIAID grants U19AI142733, U19AI135972, and DoD grant W81XWH-20-1-0270, and by the generous support of the JPB Foundation, the Open Philanthropy Project (research grant 2020-215611 (5384)), and anonymous donors to A García-Sastre. S Yildiz received funding from a Swiss National Foundation Early Postdoc Mobility fellowship (P2GEP3_184202).

## Author Contributions

JF Varona: conceptualization, investigation, and writing—original draft, review, and editing.

P Landete: investigation.

JA Lopez-Martin: conceptualization, data curation, software, formal analysis, supervision, validation, investigation, visualization, methodology, and writing—original draft, review, and editing.

V Estrada: conceptualization and investigation.

R Paredes: conceptualization and investigation.

P Guisado-Vasco: investigation.

L Fernandez de Orueta: investigation.

M Torralba: investigation.

J Fortun: conceptualization and investigation.

R Vates: investigation.

J Barberan: conceptualization and investigation.

B Clotet: investigation.

J Ancochea: investigation.

D Carnevali: investigation.

N Cabello: investigation.

L Porras: investigation.

P Gijon: investigation.

A Monereo: investigation.

D Abad: investigation.

S Zuniga: conceptualization, resources, data curation, supervision, validation, investigation, and project administration.

I Sola: conceptualization, resources, data curation, supervision, validation, investigation, and project administration.

J Rodon: conceptualization, resources, data curation, formal analysis, supervision, validation, and investigation.

J Vergara-Alert: conceptualization, data curation, and investigation.

N Izquierdo-Useros: conceptualization, resources, data curation, supervision, validation, investigation, project administration, and writing—review and editing.

S Fudio: conceptualization, data curation, software, formal analysis, visualization, and methodology.

MJ Pontes: conceptualization, data curation, and project administration.

B de Rivas: conceptualization, data curation, and project administration.

P Giron de Velasco: data curation, formal analysis, investigation, visualization, and methodology.

A Nieto: conceptualization, data curation, and formal analysis.

J Gomez: conceptualization, data curation, formal analysis, and visualization.

P Aviles: investigation.

R Lubomirov: conceptualization, data curation, software, formal analysis, visualization, and methodology.

A Belgrano: conceptualization, data curation, and formal analysis.

B Sopesen: investigation.

KM White: conceptualization, data curation, formal analysis, validation, investigation, project administration, and writing—original draft, review, and editing.

R Rosales: conceptualization, resources, supervision, validation, and investigation.

S Yildiz: conceptualization, resources, supervision, validation, and investigation.

A-K Reuschl: formal analysis and investigation.

LG Thorne: formal analysis and investigation.

C Jolly: formal analysis and investigation.

GJ Towers: formal analysis and investigation.

L Zuliani-Alvarez: formal analysis and investigation.

M Bouhaddou: formal analysis and investigation.

K Obernier: formal analysis and investigation.

BL McGovern: conceptualization, resources, supervision, validation, and investigation.

ML Rodriguez: conceptualization, resources, supervision, validation, and investigation.

L Enjuanes: conceptualization, resources, data curation, formal analysis, supervision, validation, investigation, and project administration.

JM Fernandez-Sousa: conceptualization, funding acquisition, and methodology.

NJ Krogan: conceptualization, data curation, formal analysis, validation, investigation, writing—original draft, and project administration.

JM Jimeno: conceptualization, data curation, formal analysis, supervision, investigation, methodology, project administration, and writing—original draft, review, and editing.

A Garcia-Sastre: conceptualization, data curation, formal analysis, validation, investigation, project administration, and writing—review and editing.

## Conflict of Interest Statement

V Estrada has received personal fees from Janssen, Gilead, and ViiV and grants from MSD. R Paredes has participated in Advisory Boards from Gilead, MSD, ViiV Healthcare, and Theratechnologies. M Torralba has received consulting fees as a member of Advisory Committee and honoraria and speaking fees from Gilead, Janssen, MSD, and ViiV Companies. J Fortún has participated in scientific events and received consulting or speaking fees or oral presentations from Pfizer, Gilead, MSD, Astellas, Novartis, and Roche.

J Ancochea has received fees for scientific consulting and/or speaking from Actelion, Air Liquide, Almirall, AstraZeneca, Boehringer Ingelheim, Carburos Médica, Chiesi, Faes Farma, Ferrer, GlaxoSmithKline, InterMune, Linde Healthcare, Menarini, MSD, Mundipharma, Novartis, Pfizer, Roche, Rovi, Sandoz, Takeda, and Teva. I Sola, S Zúñiga, and L Enjuanes hold a Technology Support contract with Pharmamar. N Izquierdo-Useros is inventor of a patent of Plitidepsin (EP20382821.5). The Krogan Laboratory has received research support from Vir Biotechnology and F Hoffmann-La Roche. NJ Krogan has consulting agreements with Maze Therapeutics and Interline Therapeutics, and is a shareholder of Tenaya Therapeutics. JM Fernández-Sousa is President and Founder of Pharmamar, SA (Madrid, Spain). JM Jimeno holds stocks of Pangaea Oncology, has a non-remunerated role in the Scientific Advisory Board and holds stocks of Phosplatin Therapeutics, and is a full-time employee of Pharmamar, SA (Madrid, Spain).The A García-Sastre laboratory has received research support from Pfizer, Senhwa Biosciences, Kenall Manufacturing, Avimex, Johnson & Johnson, Dynavax, 7Hills Pharma, Pharmamar, ImmunityBio, Accurius, and Nanocomposix. A García-Sastre has consulting agreements for the following companies involving cash and/or stock: Vivaldi Biosciences, Contrafect, 7Hills Pharma, Avimex, Vaxalto, Pagoda, Accurius, Esperovax, Farmak, and Pfizer. A García-Sastre is inventor on patents and patent application on the use of antivirals for the treatment of virus infections, owned by the Icahn School of Medicine at Mount Sinai, New York. A patent application based on this work has been filed (EP20382821.5). JA Lopez-Martin, S Fudio, MJ Pontes, B de Rivas, A Nieto, J Gómez, P Girón de Velasco, P Avilés, R Lubomirov, A Belgrano, and B Sopesén are employees and shareholders of Pharmamar, SA (Madrid, Spain). JA Lopez-Martin is a co-inventor of a patent for plitidepsin (WO2008135793A1). JM Jimeno is a co-inventor on a patent for didmenin (WO99/42125) and on patents for aplidine (WO03/033013 and WO 2004/080421).

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
