## [Reviewer comments · Life Science Alliance]

Pre-clinical and randomized phase I studies of plitidepsin in adults hospitalized with COVID-19

Jose Varona, Pedro Landete, Jose A. Lopez-Martin, Vicente Estrada, Roger Paredes, Pablo Guisado-Vasco, Lucia Fernandez de Orueta, Miguel Torralba, Jesus Fortun, Roberto Vates, Jose Barberan, Bonaventura Clotet, Julio Ancochea, Daniel Carnevali, Noemi Cabello, Lourdes Porras, Paloma Gijon, Alfonso Monereo, Daniel Abad, Sonia Zuñiga, Isabel Sola, Jordi Rodon, Júlia Vergara-Alert, Nuria Izquierdo-Useros, Salvador Fudio, Maria Jose Pontes, Beatriz de Rivas, Patricia Giron de Velasco, Antonio Nieto, Javier Gomez, Pablo Aviles, Rubin Lubomirov, Alvaro Belgrano, Belen Sopesen, Kris White, Romel Rosales, Soner Yildiz, Ann-Kathrin Reuschl, Lucy Thorne, Clare Jolly, Greg Towers, Lorena Zuliani-Alvarez, Mehdi Bouhaddou, Kirsten Obernier, Briana McGovern, M Luis Rodriguez, Luis Enjuanes, Jose M Fernandez-Sousa, Nevan Krogan, Jose M Jimeno, and Adolfo Garcia-Sastre

DOI: <https://doi.org/10.26508/lsa.202101200>

Corresponding author(s): Jose Varona, Departamento de Medicina Interna, Hospital Universitario HM Montepincipe, HM Hospitales, Madrid, Spain & Facultad de Medicina, Universidad San Pablo-CEU, Madrid, Spain.

Review Timeline:

Submission Date:	2021-08-18
Editorial Decision:	2021-10-22
Revision Received:	2021-11-04
Editorial Decision:	2021-11-24
Revision Received:	2021-12-15
Editorial Decision:	2021-12-20
Revision Received:	2021-12-24
Accepted:	2021-12-28

Transaction Report:

October 22, 2021

Re: Life Science Alliance manuscript #LSA-2021-01200-T

Dr. Jose F Varona
Departamento de Medicina Interna, Hospital Universitario HM Montepíncipe, HM Hospitales
Facultad de Medicina, Universidad San Pablo-CEU
Madrid, Spain

Dear Dr. Varona,

Thank you for submitting your manuscript entitled "Plitidepsin has a positive therapeutic index in adult patients with COVID-19 requiring hospitalization" to Life Science Alliance. The manuscript was assessed by expert reviewers, whose comments are appended to this letter. We invite you to submit a revised manuscript addressing the Reviewer comments.

Thank you for this interesting contribution to Life Science Alliance. We are looking forward to receiving your revised manuscript.

Sincerely,

B. MANUSCRIPT ORGANIZATION AND FORMATTING:

Reviewer #2 (Comments to the Authors (Required)):

Dear editor,

In this article, the authors provide results of pre-clinical analyses directed to evaluate inhibition of viral replication in vitro after treatment with plitidepsin of human hepatoma and kidney epithelial cell lines infected with different coronaviruses (229E, SARS-CoV, and SARS-CoV-2), as well as different strains of SARS-CoV-2 in human respiratory and gastrointestinal cell lines, showing that viral inhibition with no cytotoxic effects can be achieved in vitro. They further determined plasma concentrations to be reached with plitidepsin to achieve potentially therapeutic ranges in humans. Lastly, the authors report a proof-of-concept clinical trial which was conducted with the intention of evaluating safety and preliminary efficacy of three different doses of an IV infusion of plitidepsin in hospitalized patients with COVID-19.

I would like to congratulate the authors of this manuscript for their many hours of work and efforts directed towards developing a much needed therapeutic for COVID-19. This could be an important piece of work to continue clinical research evaluating plitidepsin for COVID-19. Furthermore, in vitro results of early evaluations were replicated by different study groups and there is evidence that the researchers followed other good practice recommendations during the execution of their trial. However, I have a few major concerns regarding how this manuscript is currently being presented which should be addressed before publication. Noteworthy, the pre-print version of this manuscript as well as a press release by the pharmaceutical company have included claims of efficacy and safety of this drug with significant interest from the press and people in the social media, reason why it is of utmost importance that any claims of safety and efficacy be fully justified before this manuscript can be published.

My main concern is that the conclusion is currently elaborated around effectiveness of plitidepsin for COVID-19 even when the study design was not suitable to evaluate effectiveness (this concept is often reserved for the evaluation of an intervention in real-world settings in the context of clinical trials) and, debatably, neither efficacy. Even when secondary outcomes were intended to determine preliminary efficacy of plitidepsin, there are a number of factors which may not allow to conclude preliminary efficacy from these outcomes in the absence of a group of reference (i.e. control group) for this particular disease. First, viral load peaks around the time of symptom onset and a reduction in viral load or having a negative PCR are situations expected to occur in the short term for all patients with COVID-19 due to the natural course of the disease (<https://www.pnas.org/content/118/8/e2017962118>). Second, all other 4 clinical outcomes refer to events which could also be expected to occur in a fraction of patients regardless of interventions, of which proportions are variable in different population settings and, since this is a phase I clinical trial with strict inclusion and exclusion criteria, patients included in this study are already at a lower risk of experiencing disease progression than the general population which makes the proportions of these 4 clinical outcomes being reported difficult to interpret and put into context compared to what has been described in most other studies. Thus, these outcomes cannot be used to conclude preliminary efficacy without a comparison group. Third, the use of dexamethasone which has been proven to improve outcomes in hospitalized patients with COVID-19 requiring oxygen therapy is a major confounder and it could be more appropriate to refer to plitidepsin + dexamethasone preliminary efficacy in any case. Despite this, I consider that this study was well designed to test safety and optimal dose of plitidepsin in patients with COVID-19 and I would encourage the authors to report their study in this direction rather than in the terms of efficacy or effectiveness.

Another important observation is that the authors have not provided the research protocol for their registered clinical trial. I was unable to find the research protocol accompanying the MedRxiv pre-print of this study, alongside the ClinicalTrials registry, or elsewhere in a publicly available website. It is a good practice to provide the research protocol for scrutiny not only during peer review, but also prior to the study being published (ideally) or alongside the published article. Please provide the study protocol with amendments as supplementary material. Otherwise, please provide any explanations in the manuscript to justify not doing so.

Lastly, since this trial involved randomization and parallel group allocation, the authors should consider reporting their study according to CONSORT recommendations (<https://www.bmj.com/content/340/bmj.c869>). Please provide the CONSORT checklist for peer review specifying where the descriptions for all items that apply to your study can be found.

The following comments and suggestions could be used by the authors to improve their manuscript:

TITLE

The current title does not accurately represent the whole study in my opinion. It would be useful to at least provide other key elements of the study design like identifying it as a preclinical + clinical study and its phase, already from the title.

ABSTRACT

1. Please adapt your abstract to include all essential key elements for reporting trials:

<https://journals.plos.org/plosmedicine/article?id=10.1371/journal.pmed.0050020>

2. Could the journal allow the authors to provide an abstract of more than 175 words if this is necessary to include all key elements?

INTRODUCTION

The current flow of ideas in the introduction is a bit confusing. The main problem as stated in the first paragraph are the COVID-19 pandemic and the lack of therapeutics. The second paragraph intends to introduce plitidepsin and its mechanism of action but fails to effectively link this idea with the preceding paragraph and its relevance for COVID-19. The third paragraph gives diffuse concepts related to plitidepsin development in human studies, and the 4th paragraph does not link the specific problem with the hypotheses and objectives of this study. I have the following suggestions to address these and other minor issues:

First paragraph: This paragraph clearly defines the main problem. I would only suggest doublechecking the dates of the first sentence with reference No. 1 since they do not match.

Second paragraph: Please separate the first sentence into individual sentences since ideas are entangled and difficult to read. Also, it could be easier to understand the flow of ideas and naturally link them with the problem stated in the preceding paragraph by reordering them into:

- 1) explain to readers what the N protein of SARS-CoV-2 is,
- 2) how the N protein interacts with eEF1A (Please also detail the nature of this interaction i.e. inhibition or inactivation?),
- 3) the role of eEF1A knockdown on viral replication (has this only been shown for viral replication or also for viral infectivity on plaque assay, for example? Also, please clarify that this has been shown for other viruses, not for SARS-CoV-2 since current references do not refer to SARS-CoV-2, otherwise, add references pertaining to SARS-CoV-2),
- 4) briefly describe plitidepsin (a cyclic depsipeptide...),
- 5) explain how plitidepsin affects eEF1A and the mechanism of this interaction (inhibition, etc),
- 6) finalize by explaining the potential relevance of plitidepsin for SARS-CoV-2 infection. Please refer to any other studies which may have hypothesized that plitidepsin could be useful in the treatment of COVID-19 or studies which have tested plitidepsin in pre-clinical or even clinical studies. This could be a good way to link the basics with clinically relevant information in the next paragraph.

Third paragraph:

- 1) Restructure the first sentence to "Plitidepsin has undergone an extensive clinical development program for the treatment of cancer".
- 2) The linking adverb "specifically" is not needed and could be eliminated.
- 3) The authors correctly give a background on plitidepsin for the treatment of cancer but should also make a statement regarding plitidepsin for the treatment of infectious diseases. If plitidepsin has never been tested for other infectious diseases, the readers should be aware of this.

Fourth paragraph: Start by explaining the specific problem that led to this paper being done. Then, specify if there were any hypotheses prior to this study being executed. End by explaining the objective of this study. To "describe the results from a Proof-of-concept trial" may be the objective of writing a paper, but it is not the objective of the study which is what we all really want to know.

RESULTS SECTION

1. Please include the relation of patients assessed for eligibility and reasons for exclusion and elimination in a flow chart as recommended by CONSORT.
2. In Figure 1 panel B description please explain what the error bars are representing (standard error, confidence interval?), as well as the dots (mean, median?). Also, why are there dots with no error bars? Is this because the error bar is too small?
3. In Figure 1 panel C, please describe IC₅₀ at the figure foot and change "{plus minus}" for "to" since a range is being represented, not an error value.
4. In the "Clinical study design and patient characteristics" section, it is advisable to also provide the links to the ClinicalTrials.gov and EudraCT registries.
5. In Table 1, are the authors not referring to races or racial groups rather than ethnic groups?
6. What is the range of altitudes for the 10 centers participating in the clinical trial? If all sites are close to sea level the cut-off value of SpO₂ 94% is appropriate, but higher altitudes could require adjustment.
7. Inclusion criteria for this study mention that RT-PCR will be performed from a nasopharyngeal swab or lower respiratory tract samples. This is important since viral load may vary depending on the sample obtained. Please clarify what samples were

- obtained from nasopharyngeal swabs or lower respiratory tract samples. This could be included in table S7. If all samples were obtained from nasopharyngeal swabs, this would not be necessary, please only clarify it in the manuscript.
8. If possible, include in Figure 3 panel A the days elapsed from symptom onset to inclusion in the study or first administration of plitidepsin (i.e. -10 to 0 days would correspond to days from symptom onset to inclusion) since this would allow to make a better interpretation of the course of disease for all patients.
 9. In Figure 3 Panel B, the x-axis appears to be incomplete. Also, there are no tags explaining what the numbers at the bottom refer to. These appear to be the number at risk, but they should be labeled. Also, it is not possible to know what the significance value refers to. Was this obtained by comparing mild vs moderate, mild vs severe, moderate vs severe? Pairwise comparisons for all these three should be provided instead and labeled.
 10. What is the justification for performing a regression analysis since there are very few (only 2) events? Also, data apparently follow a Poisson distribution rather than a binomial distribution. Why was a logistic regression applied? It looks like the data would have allowed for a Cox regression to be performed. Moreover, considering the low number of events, a Poisson regression could have been considered.
 11. In Figure 3 Panel D why are only patients with moderate COVID-19 at admission presented?
 12. In Figure 3 Panel E, it could be better to provide confidence intervals instead of the plots since it cannot be interpreted in its current form. There appears to be an error in the way the scale is being presented.
 13. In Figure 3, please describe in the figure foot the 6-point ordinal scale.
 14. In Figure 4 foot, please remove all subjective comments on the way you are interpreting the figures and graphs. Only the necessary descriptions of the images and graphs should be provided so that readers can elaborate their own unbiased interpretations.
 15. Please make sure to link all supplementary figures and tables with the results or methods sections in your manuscript. Please also describe all abbreviations in supplementary figures and tables descriptions. All figures and tables referring to the 6-point ordinal scale should describe what every number corresponds to since these cannot be readily interpreted. Also, please change "{plus minus}" for "SD" whenever standard deviation is used, or "to" for ranges since these are more appropriate terminology.
 16. In Table S7 there are patients with undetectable baseline viral load. Did these patients have any other confirmatory SARS-CoV-2 positive test? Otherwise, why were these patients included? Also, other patients had subsequent negative tests with latter positive viral loads. How do you interpret this?

DISCUSSION

1. Overall, the tone of the discussion is appropriate. Conclusions in the abstract, however, are out of tone. Please consider the major comments at the start of my review to moderate conclusions to a tone that coincides with your discussion.
2. Please discuss the limitation of having included dexamethasone as part of the amendment and how it could have influenced survival. There is uncertainty on the potential effect of dexamethasone for patients not receiving oxygen therapy since these patients under dexamethasone have had higher mortality rates than patients not receiving systemic corticosteroids, so please discuss how including dexamethasone could affect the potential applications of plitidepsin. Should plitidepsin be reserved in the future only for patients already under oxygen therapy?
3. There is evidence suggesting that systemic corticosteroids may delay viral clearance in patients with mild disease (<http://www.pnas.org/lookup/doi/10.1073/pnas.2017962118>), please also consider discussing this and its importance towards further development of plitidepsin (implications for the type of patients in whom plitidepsin should be further studied).
4. Also, please make a literature search on all other treatments that were used alongside plitidepsin (ranitidine, diphenhydramine, ondansetron), have these been hypothesized to have any potential role for SARS-CoV-2 infection and could these have any influence on clinical outcomes?
5. The first paragraph could be eliminated from this section and instead adapted or merged with the first paragraph in the introduction.
6. Reference 25 does not support this statement since it refers to potentially useful interventions for COVID-19 rather than interventions that have already proven to have efficacy. Please update this reference, consider referring to recent systematic reviews.
7. In the phrase "The death rate due to COVID-19 in our study was 6.7" what is the unit of this measure?

MATERIALS AND METHODS SECTION

1. In the "Participants" section, not all exclusion criteria have been described according to those in the trial registry.
2. In the "Randomization" section, the authors could link the sequence of enrollment to figure S2 since this figure is not referenced elsewhere in the manuscript.
3. How was the method of randomization performed? Were computer generation allocation tools used?
4. Similarly, Figure S3 could be referenced in the "Interventions" section.
5. In the "Interventions" section, please restructure this sentence since it is not clear: "In a protocol amendment dated 13 August 2020, prophylactic IV medications ondansetron 8 mg (slow infusion), diphenhydramine 25 mg, ranitidine 50 mg, and dexamethasone 8 mg (replacing oral administration) were added 30 minutes prior to plitidepsin infusion."
6. In the "Outcomes" section, no clear description of all outcomes has been given. The authors should specify the individual outcomes which were evaluated, alongside the measure used for assessment (i.e. proportion of patients who developed ...).
7. Was chest CT performed for all patients at specific follow-up times? How were chest computed tomography images

assessed? Was blinding of expert radiologists or clinicians used? Was the method or procedure of assessment of images described in the research protocol? Have the images been stored? These are important aspects to clarify since figure 4 includes a chest CT comparison for one patient.

8. The "Statistical methods" section is vague and should be expanded. Enough explanation should be given on how primary and secondary outcomes were analyzed. This section should provide enough description for someone to reproduce analyses in the hypothetical case that the dataset is requested. There should be a clear flow with enough descriptions, for example: descriptive analyses, inferential analyses, subanalyses, secondary analyses, statistical assumptions, software, and other statistical considerations (i.e. how was significance defined).

9. Kaplan-Meier analyses are not mentioned in statistical methods.

10. Regression analyses are not mentioned in statistical methods.

Referee Cross-Comments

I agree with most of the reviewer's observations. However, I think that the study is not necessarily biased but has been inadequately interpreted and reported which can easily make one think that the clinical trial was biased. The authors need to provide several clarifications and adaptations of their manuscript to substantiate. Many of these could be solved by properly reporting their manuscript according to CONSORT recommendations.

Reviewer #3 (Comments to the Authors (Required)):

In this article the authors present results about the impact of Plitidepsin (marine derived cyclic peptide) on SARS-CoV2 replication. While the in vitro results are very conclusive showing a clear decrease of SARS-CoV2 replication in presence of Plitidepsin (Figure 1) and the explanation of the pharmacological determination of Plitidepsin concentration administrated to patients is clear (Figure 2) I am scared the full design of the Human trial is biased. I thought that with the French story of hydroxychloroquine it was clear that a control group not receiving the compound tested, in this case Plitidepsin, is not optional in order to conclude about the efficiency of the drug tested.

Minor revision:

The resolution of Figure 1 panel C, Figure 3, and Figure 4 is so bad we can't read the text which is very bad for a reader and making only Figure 2 and Figure 1A and 1B readable.

Same color code (bleue, green, pink) is used for different meaning Figure 3: dose of Plitidepsin and COVID-19 severity (Panel C). Panel B is not readable at all...

Consistency is very important sometimes the Plitidepsin dose unit is mention on the figure, sometimes not. Sometimes text appear on the figure panels sometimes not.

Please take it seriously since all this mistake could let the reviewer think you are not taking this submission seriously even more when the figures are not readable.

Figure S5 and S6 are not explained properly so I couldn't judge their meaning I understand the 6-point scale and I do think Figure S6 has only 4 categories and not 6. I do think a gradient between 2 colors would have been easier to visualize if there is improvement. However, without control group we can't know if it is Plitidepsin dependent.

Major revision:

Figure 3 and 4 cannot be interpreted in the absence of a similar cohort not receiving Plitidepsin. The author always state that Plitidepsin is responsible for the improvement but we all know that the heterogeneity among COVID-19 patients in term of recovery is pretty big, so the authors have no proof without proper control to state that Plitidepsin is responsible for it.

Sadly, the results showed figure 4 can argue against Plitidepsin efficacy since there is no progressive effect of Plitidepsin increasing dose on the different parameters followed. 2mg/day could have an effect when 1.5 and 2.5mg/day won't have? is it not weird?

The manuscript is well written and easy to read.

José F. Varona MD
Hospital Universitario
HM Montepíncipe (Madrid)
Department of Internal Medicine
E-mail: jfvarona@hmhospitales.com
November 2, 2020

Eric Sawey, PhD
Life Science Alliance
Executive Editor

Dear Dr. Sawey,

On behalf of the authors, I would like to thank you for considering our manuscript previously titled “Plitidepsin has a positive therapeutic index in adult patients with COVID-19 requiring hospitalization” (LSA-2021-01200-T) that was submitted as a research article to *Life Science Alliance*.

We appreciate the opportunity to revise our manuscript in order to fully address the comments and concerns of the referees.

Below, we address their recommendations point by point.

Journal Requirements

As our submission was transferred directly from another journal, we have taken the opportunity to revise the manuscript to align with the formatting guidelines of *Life Science Alliance*.

Briefly, we have:

- Included all of the relevant Title Page information
- Adjusted the manuscript Title to conform with character counts and as per suggestions by Reviewer 2 (below)
- Edited the Summary Blurb to conform with character counts
- Revised the abstract to conform to word counts and as per suggestions by Reviewer 2 (below)
- Provided high-resolution main and supplementary figure files

Reviewer 2

We thank the reviewer for their comprehensive review of our manuscript. Below, we address their concerns point by point.

- *“My main concern is that the conclusion is currently elaborated around effectiveness of plitidepsin for COVID-19 even when the study design was not suitable to evaluate effectiveness (this concept is often reserved for the evaluation of an intervention in real-world settings in the context of clinical trials) and, debatably, neither efficacy. Even when secondary outcomes were intended to determine preliminary efficacy of plitidepsin, there are a number of factors which may not allow to conclude preliminary efficacy from these outcomes in the absence of a group of reference (i.e., control group) for this particular disease.*

First, viral load peaks around the time of symptom onset and a reduction in viral load or having a negative PCR are situations expected to occur in the short term for all patients with COVID-19 due to the natural course of the disease.

Second, all other 4 clinical outcomes refer to events which could also be expected to occur in a fraction of patients regardless of interventions, of which proportions are variable in different population settings and, since this is a phase I clinical trial with strict inclusion and exclusion criteria, patients included in this study are already at a lower risk of experiencing disease progression than the general population which makes the proportions of these 4 clinical outcomes being reported difficult to interpret and put into context compared to what has been described in most other studies. Thus, these outcomes cannot be used to conclude preliminary efficacy without a comparison group.

Third, the use of dexamethasone which has been proven to improve outcomes in hospitalized patients with COVID-19 requiring oxygen therapy is a major confounder and it could be more appropriate to refer to plitidepsin + dexamethasone preliminary efficacy in any case.

Despite this, I consider that this study was well designed to test safety and optimal dose of plitidepsin in patients with COVID-19 and I would encourage the authors to report their study in this direction rather than in the terms of efficacy or effectiveness.”

We agree with the reviewer that the main thrust of this research has been to evaluate the safety/tolerability and appropriate dose of plitidepsin in hospitalized patients with COVID-19, and that all efficacy end points were secondary to that aim. We have therefore revised the manuscript to reinforce that this was, first and foremost, a safety study on the use of plitidepsin for the treatment of infectious disease. Although we continue to include the secondary efficacy endpoints, we agree with the reviewer that no claims of causality can be made without a well-designed, randomized, controlled clinical trial.

As noted by the reviewer, we have misused the word ‘effectiveness’ when ‘efficacy’ is the appropriate term for this study. This has been changed throughout the manuscript. Furthermore, where necessary, we have ensured that all conclusions in this manuscript refer to ‘preliminary efficacy’, given the limitations of this phase 1 study.

- *“Another important observation is that the authors have not provided the research protocol for their registered clinical trial. I was unable to find the research protocol accompanying the MedRxiv pre-print of this study, alongside the ClinicalTrials registry, or elsewhere in a publicly available website. It is a good practice to provide the research protocol for scrutiny not only during peer review, but also prior to the study being published (ideally) or alongside the published article. Please provide the study protocol with amendments as supplementary material. Otherwise, please provide any explanations in the manuscript to justify not doing so.”*

We agree with the reviewer that the study protocol should be made available to scrutiny. As this manuscript was transferred directly from another journal, the study protocol was not included with the original submission. We have included the full study protocol with this revised submission to *Life Science Alliance*.

- *“Lastly, since this trial involved randomization and parallel group allocation, the authors should consider reporting their study according to CONSORT recommendations (<https://www.bmj.com/content/340/bmj.c869>). Please provide the CONSORT checklist for peer review specifying where the descriptions for all items that apply to your study can be found.”*

We thank the reviewer for this recommendation and have included the CONSORT checklist with this revised submission.

- *“The following comments and suggestions could be used by the authors to improve their manuscript:*

TITLE

The current title does not accurately represent the whole study in my opinion. It would be useful to at least provide other key elements of the study design like identifying it as a preclinical + clinical study and its phase, already from the title.”

We thank the reviewer for this recommendation and have included the fact that this is a pre-clinical and phase I study in the title.

- **“ABSTRACT**

*Please adapt your abstract to include all essential key elements for reporting trials:
<https://journals.plos.org/plosmedicine/article?id=10.1371/journal.pmed.0050020>
Could the journal allow the authors to provide an abstract of more than 175 words if this is necessary to include all key elements?”*

We thank the reviewer for this suggestion and have aligned the abstract to the CONSORT guidance as recommended. The current abstract contains all of the key elements, and at 182 words, is only slightly over the 175-word limit for abstracts.

- **“INTRODUCTION**

*The current flow of ideas in the introduction is a bit confusing.
I have the following suggestions to address these and other minor issues:
First paragraph: This paragraph clearly defines the main problem. I would only suggest doublechecking the dates of the first sentence with reference No. 1 since they do not match.”*

We have updated the dates and statistics of paragraph 1, as COVID-19 has continued to spread since our original submission.

- *“Second paragraph: Please separate the first sentence into individual sentences since ideas are entangled and difficult to read. Also, it could be easier to understand the flow of ideas and naturally link them with the problem stated in the preceding paragraph by reordering them into:
explain to readers what the N protein of SARS-CoV-2 is,
how the N protein interacts with eEF1A (Please also detail the nature of this interaction i.e. inhibition or inactivation?),
the role of eEF1A knockdown on viral replication (has this only been shown for viral replication or also for viral infectivity on plaque assay, for example? Also, please clarify that this has been shown for other viruses, not for SARS-CoV-2 since current references do not refer to SARS-CoV-2, otherwise, add references pertaining to SARS-CoV-2),
briefly describe plitidepsin (a cyclic depsipeptide...),
explain how plitidepsin affects eEF1A and the mechanism of this interaction (inhibition, etc),
finalize by explaining the potential relevance of plitidepsin for SARS-CoV-2 infection.*

Please refer to any other studies which may have hypothesized that plitidepsin could be useful in the treatment of COVID-19 or studies which have tested plitidepsin in pre-clinical or even clinical studies. This could be a good way to link the basics with clinically relevant information in the next paragraph.”

We thank the reviewer for these suggestions and have followed the flow that has been recommended. With regard to some of the reviewer’s specific points:

- The interaction between viral proteins, like the SARS-CoV-2 N protein, and eEF1A is neither inhibitory nor inactivation. Instead eEF1A is an important host factor for the replication of viral pathogens. This has been clarified in the current manuscript.
- Downregulation or chemical inhibition of eEF1A has been shown to impair both viral replication and infectivity, a point that has been clarified in the manuscript. We have also added references supporting this activity in SARS-CoV-2, as well as other viruses.
- *“Third paragraph:
Restructure the first sentence to "Plitidepsin has undergone an extensive clinical development program for the treatment of cancer".
The linking adverb "specifically" is not needed and could be eliminated.
The authors correctly give a background on plitidepsin for the treatment of cancer but should also make a statement regarding plitidepsin for the treatment of infectious diseases. If plitidepsin has never been tested for other infectious diseases, the readers should be aware of this.”*

We thank the reviewer for these suggestions and have followed the flow that has been recommended. Given that this paragraph currently discusses the previous clinical development for cancer, we have moved the final suggestion on plitidepsin’s evaluation in infectious disease to the fourth paragraph.

- *“Fourth paragraph: Start by explaining the specific problem that led to this paper being done. Then, specify if there were any hypotheses prior to this study being executed. End by explaining the objective of this study. To "describe the results from a Proof-of-concept trial" may be the objective of writing a paper, but it is not the objective of the study which is what we all really want to know.”*

We thank the reviewer for these suggestions and have clarified that the purpose of this work was to evaluate the safety and tolerability of plitidepsin across three dose levels in patients hospitalized with COVID-19. We also note that preliminary efficacy endpoints were also evaluated.

- **RESULTS SECTION**
1. Please include the relation of patients assessed for eligibility and reasons for exclusion and elimination in a flow chart as recommended by CONSORT.

We have provided a CONSORT flow chart on patient eligibility and follow-up as Figure 4 of the revised manuscript.

- *2. In Figure 1 panel B description please explain what the error bars are representing (standard error, confidence interval?), as well as the dots (mean, median?). Also, why are there dots with no error bars? Is this because the error bar is too small?*

We thank the reviewer for this observation and have revised the figure axis to explain that the dots represent mean cytotoxicity. Additionally, error bars represent standard deviation and dots without error bars are because the error bars are too small (both have been noted in the revised figure legend).

- 3. In Figure 1 panel C, please describe IC50 at the figure foot and change "{plus minus}" for "to" since a range is being represented, not an error value.

We would like to comment that the IC50 values shown are not a range, but instead are the mean IC50 along with the standard error of the mean. This has been clarified in the figure legend.

- 4. In the "Clinical study design and patient characteristics" section, it is advisable to also provide the links to the ClinicalTrials.gov and EudraCT registries.

We thank the reviewer for this comment. In order to maintain the flow of the manuscript, we have minimized the study design in the Results section, moving the content to the Materials and Methods. There, we have included links to the registries named.

- 5. In Table 1, are the authors not referring to races or racial groups rather than ethnic groups?

Yes, this is correct. We have revised Table 1 to refer to races rather than ethnic groups.

- 6. What is the range of altitudes for the 10 centers participating in the clinical trial? If all sites are close to sea level the cut-off value of SpO2 94% is appropriate, but higher altitudes could require adjustment.

We appreciate this insightful comment from the reviewer. The 10 participating sites were located in Spanish cities between 192–876 m above sea level (median= 684 m, IQR= [632.2, 701.8 m]).

Given that the clinical relevance of differences in oxygen saturation in normal subjects between 0 to 1500 m above the sea level has not been demonstrated [1], and that the Berlin criteria for the definition of Acute Respiratory Distress Syndrome only require the PaO2/FiO2 ratio to be adjusted at altitudes over 1000 m above the sea level [2], we believe it is unlikely that altitude had any clinical impact on the interpretations of our study and that SpO2 cut-off adjustment is not necessary.

[1] Goldberg, S., et al. (2012). "Effect of Moderate Elevation above Sea Level on Blood Oxygen Saturation in Healthy Young Adults." *Respiration* 84(3): 207-211.

[2] Ranieri, V. M., et al. (2012). "Acute respiratory distress syndrome: the Berlin Definition." *JAMA* 307(23): 2526-2533.

- 7. Inclusion criteria for this study mention that RT-PCR will be performed from a nasopharyngeal swab or lower respiratory tract samples. This is important since viral load may vary depending on the sample obtained. Please clarify what samples were obtained from nasopharyngeal swabs or lower respiratory tract samples. This could be

included in table S7. If all samples were obtained from nasopharyngeal swabs, this would not be necessary, please only clarify it in the manuscript.

Although it is correct that the protocol allowed for RT-PCR analysis of nasopharyngeal swabs or lower respiratory tract samples, all samples were ultimately obtained from nasopharyngeal swabs. This point has been noted in methods section of the manuscript.

- *8. If possible, include in Figure 3 panel A the days elapsed from symptom onset to inclusion in the study or first administration of plitidepsin (i.e. -10 to 0 days would correspond to days from symptom onset to inclusion) since this would allow to make a better interpretation of the course of disease for all patients.*

We thank the reviewer for this comment. We have updated the ‘swimmer plot’ with the symptom onset timepoint (Figure 5 panel B).

- *9. In Figure 3 Panel B, the x-axis appears to be incomplete. Also, there are no tags explaining what the numbers at the bottom refer to. These appear to be the number at risk, but they should be labeled. Also, it is not possible to know what the significance value refers to. Was this obtained by comparing mild vs moderate, mild vs severe, moderate vs severe? Pairwise comparisons for all these three should be provided instead and labeled.*

Yes, we see that the x-axis has been truncated and have corrected the figure. We have also labeled the bottom numbers as number at risk, and have deleted the p-value as this was not an objective of the protocol.

- *10. What is the justification for performing a regression analysis since there are very few (only 2) events? Also, data apparently follow a Poisson distribution rather than a binomial distribution. Why was a logistic regression applied? It looks like the data would have allowed for a Cox regression to be performed. Moreover, considering the low number of events, a Poisson regression could have been considered.*

We thank the reviewer for this comment, and note that our rationale was to predict the status of being discharged and alive at day 8, which is dichotomic, and has a frequency of 56.8%. Therefore, we believe that regression models are appropriate. Nevertheless, as this was a *post-hoc* exploratory exercise, we have decided to omit this analysis from the revised manuscript.

- *11. In Figure 3 Panel D why are only patients with moderate COVID-19 at admission presented?*

We thank the reviewer for the opportunity to clarify. Given that most of patients in this study presented with moderate COVID-19 at baseline (23 of 45; 51%), and that they were equally distributed amongst the dosing levels (8 received 1.5 mg plitidepsin/day, 7 received 2.0 mg/day, and 8 received 2.5 mg/day), we sought to analyze their outcomes by dose as visualized in the panel in question. Furthermore, patients with moderate COVID-19 may represent the potential target population for further development of plitidepsin as a COVID-19 therapy, so this exploratory analysis serves as hypothesis generation for any future phase III clinical trials, a point that has been made explicit in the revised manuscript.

- *12. In Figure 3 Panel E, it could be better to provide confidence intervals instead of the plots since it cannot be interpreted in its current form. There appears to be an error in the way the scale is being presented.*

Yes, we see that this figure has not been reproduced correctly and have decided that, since this is an exploratory analysis, it does not merit inclusion in the revised manuscript and has been removed.

- *13. In Figure 3, please describe in the figure foot the 6-point ordinal scale.*

In each of the figures where the 6-point ordinal scale is described we have added a brief description of it to the figure legend, in addition to the citation of the source reference.

- *14. In Figure 4 foot, please remove all subjective comments on the way you are interpreting the figures and graphs. Only the necessary descriptions of the images and graphs should be provided so that readers can elaborate their own unbiased interpretations.*

We thank the reviewer for this suggestion and have removed all commentary from the figure legend, allowing the data to speak for themselves.

- *15. Please make sure to link all supplementary figures and tables with the results or methods sections in your manuscript. Please also describe all abbreviations in supplementary figures and tables descriptions. All figures and tables referring to the 6-point ordinal scale should describe what every number corresponds to since these cannot be readily interpreted. Also, please change "{plus minus}" for "SD" whenever standard deviation is used, or "to" for ranges since these are more appropriate terminology.*

We have revised the manuscript to ensure that all the supplementary materials are correctly linked with sections in the body of the manuscript. Additionally, we have ensured that all abbreviations in the supplementary materials are defined, that all mention of the 6-point ordinal scale describes the scale for ease of interpretability, and that ‘standard deviation (SD)’ and ‘to’ are used appropriately throughout the revised manuscript.

- *16. In Table S7 there are patients with undetectable baseline viral load. Did these patients have any other confirmatory SARS-CoV-2 positive test? Otherwise, why were these patients included? Also, other patients had subsequent negative tests with latter positive viral loads. How do you interpret this?*

We thank the reviewer for the chance to clarify this point. All participants had a positive PCR test for SARS-CoV-2, performed in an authorized laboratory (all sites were located in Spain, where Health Authorities were responsible for this authorization). This was one of the criteria for eligibility, and not the quantitative RT-PCR result, which, on the other hand, was not immediate, and was instead outsourced to a third party central laboratory. The methodology and sensitivity of this test has been detailed in the revised manuscript. Potential explanations for both situations (undetectable viral load at baseline and fluctuations in the detection) could be as follows:

- Inadequate sampling: Unfortunately sampling of biological material can be heterogeneous and, for nasopharyngeal swabs, the site of sampling is not visible.
- Rapid clearance of virus in the nasopharyngeal space, after initial diagnosis. This has been described even in patients progressing to severe disease.
- Suboptimal sensitivity of the assay - we acknowledge this as a limitation, but note that this kit was among the best performing at the time the trial was conducted.

○ **DISCUSSION**

1. Overall, the tone of the discussion is appropriate. Conclusions in the abstract, however, are out of tone. Please consider the major comments at the start of my review to moderate conclusions to a tone that coincides with your discussion.

We thank the reviewer for this comment. We have revised the discussion in line with the comments made previously with regard to this study being a Phase I trial, primarily focused on safety, and have reduced the focus on our preliminary efficacy data.

- *2. Please discuss the limitation of having included dexamethasone as part of the amendment and how it could have influenced survival. There is uncertainty on the potential effect of dexamethasone for patients not receiving oxygen therapy since these patients under dexamethasone have had higher mortality rates than patients not receiving systemic corticosteroids, so please discuss how including dexamethasone could affect the potential applications of plitidepsin. Should plitidepsin be reserved in the future only for patients already under oxygen therapy?*
- *3. There is evidence suggesting that systemic corticosteroids may delay viral clearance in patients with mild disease (<http://www.pnas.org/lookup/doi/10.1073/pnas.2017962118>), please also consider discussing this and its importance towards further development of plitidepsin (implications for the type of patients in whom plitidepsin should be further studied).*

We appreciate the reviewer's comments and wish to highlight that the amendment changed only the route of administration of dexamethasone from oral to IV, given the latter's more rapid onset of action. Therefore, all patients received dexamethasone 8 mg/day for a minimum of three days.

In addition, we have described in the manuscript that 64.4% of the patients received glucocorticoids for other purposes outside of COVID-19 treatment, beyond the 3 day plitidepsin treatment period.

With regard to considering plitidepsin only for patients under oxygen therapy, this is in line with the design of the currently ongoing phase III trial. That said, the reason for that design was to minimize the variability in co-medications patients are receiving. Admitting that this strategy has fewer confounding factors is not the same, in our opinion, as to state that plitidepsin should be reserved only for patients under oxygen therapy.

We would also like to point out that all patients received dexamethasone, which is known to induce lymphopenia, and it is noteworthy that this study showed a median increase in the absolute number of lymphocytes. We believe that this is a very interesting finding, given that SARS-CoV-2 infection can induce lymphopenia that may be responsible for delaying immune responses. Additionally, this lymphopenia has been also associated with the interaction between the SARS-CoV N viral protein and host EF1A.

(Ref: Zhou B, Liu J, Wang Q, et al. The nucleocapsid protein of severe acute respiratory syndrome coronavirus inhibits cell cytokinesis and proliferation by interacting with translation elongation factor 1alpha. *J Virol.* 2008;82(14):6962-6971. doi:10.1128/JVI.00133-08)

- 4. Also, please make a literature search on all other treatments that were used alongside plitidepsin (ranitidine, diphenhydramine, ondansertron), have these been hypothesized to have any potential role for SARS-CoV-2 infection and could these have any influence on clinical outcomes?

This is an interesting consideration that can only be fully addressed with results from a randomized controlled clinical study.

From the list of drugs used as a premedication, ranitidine (and famotidine) is the one with the highest abundance of data. Nevertheless, an expert panel from the IDSA (Infectious Diseases Society of America) determined that, "the certainty of evidence to be very low due to concerns with risk of bias, imprecision, and possible publication bias. The panel agreed that critically ill patients (i.e., mechanically ventilated) may have been more likely to receive PPIs than famotidine, thus potentially allocating more prognostically favorable patients to the famotidine group; however, the study did not report a protective effect associated with the use of PPIs."

[1] Yuan, S., Wang, R., Chan, J.FW. et al. Metallo drug ranitidine bismuth citrate suppresses SARS-CoV-2 replication and relieves virus-associated pneumonia in Syrian hamsters. *Nat Microbiol* 5, 1439–1448 (2020). <https://doi.org/10.1038/s41564-020-00802-x>

[2] Freedberg DE, Conigliaro J, Wang TC, Tracey KJ, Callahan MV, Abrams JA; Famotidine Research Group. Famotidine Use Is Associated With Improved Clinical Outcomes in Hospitalized COVID-19 Patients: A Propensity Score Matched Retrospective Cohort Study. *Gastroenterology.* 2020 Sep;159(3):1129-1131.e3. doi: 10.1053/j.gastro.2020.05.053. Epub 2020 May 22. PMID: 32446698; PMCID: PMC7242191.

[3] Bhimraj A, Morgan RL, Shumaker AH, Lavergne V, Baden L, Cheng VC, Edwards KM, Gandhi R, Gallagher J, Muller WJ, O'Horo JC, Shoham S, Murad MH, Mustafa RA, Sultan S, Falck-Ytter Y. Infectious Diseases Society of America Guidelines on the Treatment and Management of Patients with COVID-19. *Infectious Diseases Society of America* 2021; Version 5.5.1. Available at <https://www.idsociety.org/practice-guideline/covid-19-guideline-treatment-and-management/> . Accessed 04-NOV-2021.

- 5. The first paragraph could be eliminated from this section and instead adapted or merged with the first paragraph in the introduction.

We agree with the reviewer, and have edited this paragraph into the introduction.

- 6. Reference 25 does not support this statement since it refers to potentially useful interventions for COVID-19 rather than interventions that have already proven to have efficacy. Please update this reference, consider referring to recent systematic reviews.

We thank the reviewer for this suggestion, and have changed the reference accordingly.

Juul S, Nielsen EE, Feinberg J, Siddiqui F, Jørgensen CK, Barot E, et al. Interventions for treatment of COVID-19: Second edition of a living systematic review with meta-analyses and trial sequential analyses (The LIVING Project). *PLoS One.* 2021;16(3):e0248132.

- 7. In the phrase "The death rate due to COVID-19 in our study was 6.7" what is the unit of this measure?

The unit of measure is percent (%) and has been added back to the manuscript.

- **MATERIALS AND METHODS SECTION**
 1. In the "Participants" section, not all exclusion criteria have been described according to those in the trial registry.

Yes, we have elected to summarize the key exclusion criteria rather than listing them all. Full exclusion criteria can be found in Supplementary Appendix 2, which includes the complete study protocol.

- 2. In the "Randomization" section, the authors could link the sequence of enrollment to figure S2 since this figure is not referenced elsewhere in the manuscript.

We thank the reviewer for this suggestion. With the addition of the CONSORT diagram, we believe that this figure is most appropriate to reference to the 'Randomization' section. Also, we have ensured that all figures in the manuscript and supplementary materials are correctly linked/referenced within the body of the manuscript itself.

- 3. How was the method of randomization performed? Were computer generation allocation tools used?

Central randomization was performed in the cohorts where it was applicable, according to a pre-defined patient enrolment procedure and treatment assignment specifications.

Once the patients were first registered in the eCRF system (OpenClinica), the system assigned a unique randomization number and the dosing group was available to the investigational sites.

The randomization lists (initial and study expansion additional list) used to feed the eCRF randomization system were created using SAS v9.4. We have ensured that these points are captured in the revised manuscript.

- 4. Similarly, Figure S3 could be referenced in the "Interventions" section.

Thank you. We have linked the diagram of treatments with the 'Interventions' section.

- 5. In the "Interventions" section, please restructure this sentence since it is not clear: "In a protocol amendment dated 13 August 2020, prophylactic IV medications ondansetron 8 mg (slow infusion), diphenhydramine 25 mg, ranitidine 50 mg, and dexamethasone 8 mg (replacing oral administration) were added 30 minutes prior to plitidepsin infusion."

We have revised this sentence, splitting it into two shorter sentences for clarity.

- *6. In the "Outcomes" section, no clear description of all outcomes has been given. The authors should specify the individual outcomes which were evaluated, alongside the measure used for assessment (i.e. proportion of patients who developed ...).*

We have elaborated in this section on which outcomes were measured in terms of proportions of patients and which ones were changes from baseline.

- *7. Was chest CT performed for all patients at specific follow-up times? How were chest computed tomography images assessed? Was blinding of expert radiologists or clinicians used? Was the method or procedure of assessment of images described in the research protocol? Have the images been stored? These are important aspects to clarify since figure 4 includes a chest CT comparison for one patient.*

We thank the reviewer for raising this point. Chest imaging was not required per protocol, and the CT images shown in the manuscript represent a relevant clinical finding from a patient whose radiology evaluation was medically indicated. To prevent confusion, these images have been moved to the supplemental appendix and their presentation is found on the new section on post-hoc analyses.

- *8. The "Statistical methods" section is vague and should be expanded. Enough explanation should be given on how primary and secondary outcomes were analyzed. This section should provide enough description for someone to reproduce analyses in the hypothetical case that the dataset is requested. There should be a clear flow with enough descriptions, for example: descriptive analyses, inferential analyses, subanalyses, secondary analyses, statistical assumptions, software, and other statistical considerations (i.e. how was significance defined).*

We have included, in the section of statistical methods, additional and mostly descriptive text on the methods. As we have made explicit in the revised manuscript, the main purpose of the study was to evaluate the safety of the 3 dosing groups. Given the exploratory nature of the efficacy analyses, no formal statistical hypothesis testing or inference was pre-planned, and the results will help to generate hypothesis to be tested in further studies.

- *9. Kaplan-Meier analyses are not mentioned in statistical methods.*
10. Regression analyses are not mentioned in statistical methods.

We thank the reviewer for this observation. We have included text on Kaplan-Meier analyses for time-to-event endpoints, as well as text on the exploratory logistic and Cox regressions that were performed in the section of Statistical Methods.

- *Referee Cross-Comments*
I agree with most of the reviewer's observations. However, I think that the study is not necessarily biased but has been inadequately interpreted and reported which can easily make one think that the clinical trial was biased. The authors need to provide several clarifications and adaptations of their manuscript to substantiate. Many of these could be solved by properly reporting their manuscript according to CONSORT recommendations.

We appreciate the reviewer's defense of our work and have taken into account all of their recommendations/suggestions in order to eliminate the impression of bias. As part of this effort,

we have revised the manuscript to focus on the primary endpoint and minimized our speculation with regard to secondary efficacy endpoints. Finally, by aligning the manuscript to the CONSORT recommendations, and disclosing all sources of funding associated with this work, we hope to be fully transparent in the purpose and findings of this study.

Reviewer #3 (Comments to the Authors (Required)):

- *In this article the authors present results about the impact of Plitidepsin (marine derived cyclic peptide) on SARS-CoV2 replication. While the in vitro results are very conclusive showing a clear decrease of SARS-CoV2 replication in presence of Plitidepsin (Figure 1) and the explanation of the pharmacological determination of Plitidepsin concentration administrated to patients is clear (Figure 2) I am scared the full design of the Human trial is biased. I thought that with the French story of hydroxychloroquine it was clear that a control group not receiving the compound tested, in this case Plitidepsin, is not optional in order to conclude about the efficiency of the drug tested.*

We thank the reviewer for their thoughtful comments, and appreciate their concern with regard to drawing efficacy conclusions from an uncontrolled phase I clinical study. As the main thrust of this research has been to evaluate the safety/tolerability and appropriate dose of plitidepsin in hospitalized patients with COVID-19, we recognize that all efficacy end points were secondary to that aim. We have therefore revised the manuscript to reinforce that this was, first and foremost, a safety study on the use of plitidepsin for the treatment of infectious disease. Although we continue to include the secondary efficacy endpoints, we agree with the reviewer that no claims of causality can be made without a well-designed, randomized, controlled clinical trial.

- *Minor revision: The resolution of Figure 1 panel C, Figure 3, and Figure 4 is so bad we can't read the text which is very bad for a reader and making only Figure 2 and Figure 1A and 1B readable. Same color code (bleue, green, pink) is used for different meaning Figure 3: dose of Plitidepsin and COVID-19 severity (Panel C). Panel B is not readable at all... Consistency is very important sometimes the Plitidepsin dose unit is mention on the figure, sometimes not. Sometimes text appear on the figure panels sometimes not. Please take it seriously since all this mistake could let the reviewer think you are not taking this submission seriously even more when the figures are not readable.*

We thank the reviewer for these comments and have replaced the figures with high-resolution images for better readability. We have also revised all figures to ensure that they are as consistent as possible.

- *Figure S5 and S6 are not explained properly so I couldn't judge their meaning I understand the 6-point scale and I do think Figure S6 has only 4 categories and not 6. I do think a gradient between 2 colors would have been easier to visualize if there is improvement. However, without control group we can't know if it is Plitidepsin dependent.*

In the revised manuscript we have provided a description in the caption of each figure highlighting the 6-point scale, so that the meaning of the scale is clear. We have also revised the colors of these images so that they scale from light to dark, so that their meaning is more obvious at a glance. Also, we agree that it is impossible to draw causal conclusions from this

study without a control group, and have modified our language when discussing the efficacy endpoints to note that much of this is preliminary and hypothesis generating for a phase III clinical study.

- *Major revision:
Figure 3 and 4 cannot be interpreted in the absence of a similar cohort not receiving Plitidepsin. The author always state that Plitidepsin is responsible for the improvement but we all know that the heterogeneity among COVID-19 patients in term of recovery is pretty big, so the authors have no proof without proper control to state that Plitidepsin is responsible for it.*

As noted above, we are in agreement with the reviewer that claims of causality cannot be made in the absence of a larger, randomized, controlled clinical study. We have moderated our language in the revised manuscript when discussing efficacy, and focused more on the important safety data that this study was designed to assess.

- *Sadly, the results showed figure 4 can argue against Plitidepsin efficacy since there is no progressive effect of Plitidepsin increasing dose on the different parameters followed. 2mg/day could have an effect when 1.5 and 2.5mg/day won't have? is it not weird?*

We believe that the absence of a clear dose response stems from the relatively small sample of patients that were assessed in this study. In the revised manuscript, we highlight the lack of an observed dose response in the limitations section of the Discussion.

Again, we would like to express our gratitude at the reviewers' thoughtful comments, and hope that our responses and improvements have addressed their concerns. We would like to note that in taking into account the reviewers' comments, we have reordered the figures, moving some to the supplement, and that the references have also been updated according to the revisions that we have made to the manuscript.

If there are any further questions or comments after this revision, please do not hesitate to contact us.

Sincerely,

José F. Varona MD

Hospital Universitario HM Montepíncipe (Madrid)
Department of Internal Medicine
Corresponding author.
E-mail: jfvarona@hmhospitales.com.

November 24, 2021

Re: Life Science Alliance manuscript #LSA-2021-01200-TR

Dr. Jose F Varona
Departamento de Medicina Interna
Hospital Universitario HM Montepíncipe
HM Hospitales
Madrid, Spain

Dear Dr. Varona,

Thank you for submitting your revised manuscript entitled "Pre-clinical and randomized phase I studies of plitidepsin in adults hospitalized with COVID-19" to Life Science Alliance. The manuscript has been seen by the original reviewers whose comments are appended below. While the reviewers continue to be overall positive about the work in terms of its suitability for Life Science Alliance, some important issues remain.

Our general policy is that papers are considered through only one revision cycle; however, given that the suggested changes are relatively minor, we are open to one additional short round of revision. Please note that I will expect to make a final decision without additional reviewer input upon resubmission.

Please submit the final revision within one month, along with a letter that includes a point by point response to the remaining reviewer comments.

To upload the revised version of your manuscript, please log in to your account: <https://lsa.msubmit.net/cgi-bin/main.plex>
You will be guided to complete the submission of your revised manuscript and to fill in all necessary information.

B. MANUSCRIPT ORGANIZATION AND FORMATTING:

Sincerely,

Reviewer #2 (Comments to the Authors (Required)):

In this resubmission, the authors have addressed some of the main comments I had made on their first version of their

manuscript, while also making significant efforts to adequately report their manuscript according to CONSORT recommendations. However, they did not adequately address all reviewers' main comments. Also, unsolicited modifications were made, and new results have been added without providing any explanations, as well as the inclusion of one more author (which I am unsure if is allowed by the journal). Therefore, this resubmission required a full re-evaluation and major and minor revisions are needed.

Major comments:

1. On the first version of the manuscript, both reviewers argued that efficacy could not be concluded from the authors' results. The authors have not properly addressed our comments. While they mention that they now focus more on safety rather than efficacy endpoints, they still make claims of efficacy and there is no contextualization of their results according to the natural course of the disease as I had previously commented on; it feels like our arguments were simply ignored. For instance, the following sentences are evidence of the authors claiming efficacy which is not supported by their study: "Patients treated with plitidepsin showed rapid reduction in viral load (compared to their baseline value), analytical improvement of biomarkers associated to inflammatory processes (Figures S4 and S5), and there is evidence of prompt clearance of pneumonia infiltrates in some participants with available chest imaging performed for medical reasons (i.e. not per protocol) (Figure S8A-B). Each of these outcomes very likely contributed to mitigating disease progression and leading to an earlier discharge from the hospital". Please delete all claims of efficacy throughout the manuscript since they are not supported by this study.
2. While it is appropriate to report all outcomes as originally described in the study registry and study protocol, the authors must make a greater effort to fully explain the meaning of their results by commenting the following points in the discussion section, which were also left ignored in their revision:
 - 2.1. Several studies have shown that viral load peaks at symptom onset with subsequent decline in viral load. This is due to the natural course of the disease, reason why viral load decline in this study cannot be used to conclude preliminary efficacy since there was no control group. References:
 - 2.1.1. Néant N, Lingas G, Le Hingrat Q, Ghosn J, Engelmann I, Lepiller Q, et al. Modeling SARS-CoV-2 viral kinetics and association with mortality in hospitalized patients from the French COVID cohort. *Proc Natl Acad Sci*. 2021 Feb 23;118(8):e2017962118.
 - 2.1.2. He X, Lau EHY, Wu P, Deng X, Wang J, Hao X, et al. Temporal dynamics in viral shedding and transmissibility of COVID-19. *Nat Med*. 2020 May 15;26(5):672-5.
 - 2.2. For the same reason, negativization of RT-PCR cannot be used to conclude preliminary efficacy in this study in the absence of a control group.
 - 2.3. This study was not suitable to evaluate preliminary efficacy on mortality nor oxygen therapy modalities since a control group was not included. Strict inclusion and exclusion criteria due to this being a phase I study limit the authors' ability to compare their findings to what has been reported in the literature. The authors failed to acknowledge these points in the limitations of their study.
3. The authors mention in the manuscript that hospital discharge was a secondary outcome, however this endpoint was not described in the trial registry, nor the trial protocol provided. It is therefore not acceptable to present their results in terms of hospital discharge rates. Outcomes need to be presented exactly as described in the study registry and protocol throughout the entire manuscript, including the abstract. All results of discharge rates could be moved to the post-hoc analysis section while also commenting in the discussion section that discharge rates were not a pre-defined endpoint which warrants caution for the interpretation of these results.
4. Improvement of inflammatory biomarkers and radiological improvements were also not primary, secondary, or exploratory endpoints. Thus, the authors cannot use these results to argue efficacy since the trial was not designed to address these hypotheses. All comments on improvements in radiological and inflammation parameters need to be omitted in the discussion section, otherwise the authors need to include the following statement within the discussion: "Changes in inflammatory biomarkers and radiological studies were not pre-defined endpoints in this trial. Thus, this study was not designed to evaluate if plitidepsin can improve these parameters and future studies could be designed to address these hypotheses".
5. Version 7.0 of the trial protocol has been provided by the authors as requested. However, protocol amendments for all prior versions are not available for scrutiny. These need to be provided within the protocol as a subheading. See SPIRIT recommendations (specifically item 25) which explain why key protocol amendments for all versions need to be described: <https://www.spirit-statement.org/protocol-amendments/>
6. The administration of other drugs (dexamethasone, diphenhydramine, ranitidine, and ondansetron) could be considered a major deviation from the international study registry since the use of these pharmaceuticals was unfortunately not described at all at any moment within the ClinicalTrials.gov registry. Since the authors have not provided information on protocol version amendments, it is impossible to corroborate exactly when and why co-interventions were implemented, which is one of the main reasons why providing all amendments to the protocol is very important for this study.
 - 6.1. Moving Supplementary Figure 1 to the main manuscript would also allow readers to have a clearer view of how the intervention was applied. Please consider having this figure in the main manuscript instead of supplementary material.
7. I had asked the authors to include in the discussion of their manuscript the potential implications of having combined dexamethasone with plitidepsin, which was also ignored. Namely, they should discuss (explicitly in the discussion section of the manuscript):
 - 1) Acknowledge that dexamethasone may be a major confounder since it appears to be the case that all patients received plitidepsin alongside dexamethasone as the authors declared in their rebuttal.
 - 2) In order to address this confounding, what kind of trial designs will be needed in the future (i.e., factorial randomized controlled trials)? This is important to discuss since we need to understand what may really work on patients, not only seek to approve drug combinations without knowing which components may or may not work.
 - 3) What kind of patients (according to disease severity) could likely benefit from the

combination of plitidepsin and dexamethasone and could some patients be harmed (i.e., mild COVID-19)? See the following reference for any clarifications of what I meant in the first round of review: Rochweg B, Agarwal A, Siemieniuk R A, Agoritsas T, Lamontagne F, Askie L et al. A living WHO guideline on drugs for covid-19 BMJ 2020; 370 :m3379 doi:10.1136/bmj.m3379. 4) If the authors consider that dexamethasone is not a major confounder, well-founded counter-arguments need to be provided in the discussion section to convincingly clarify why it is not a confounder since only mentioning that the authors do not think that dexamethasone is a confounder is not acceptable. 5) There is evidence that dexamethasone can delay viral clearance in patients with mild disease (<http://www.pnas.org/lookup/doi/10.1073/pnas.2017962118>), how could this affect patients receiving dexamethasone plus plitidepsin? If the answer is "I am not sure", then this needs to be recognized in the discussion as a potential risk until studies are performed to evaluate this hypothesis. It is important to discuss all these points so that this paper can be useful to continue investigating drugs for COVID-19 in a safe way while considering all potential risks.

8. The mean 15-day viral load reduction value has been changed in the abstract without any explanations. Also, it is not clear why the only secondary outcome being presented is the result for viral clearance at day 15 since this outcome was set to be measured up to day 31. The authors should either describe all secondary outcomes and present all results for secondary outcomes in the abstract, or leave all secondary outcomes for the main manuscript. Notice that CONSORT only requires the primary outcome to be presented from the abstract.

9. The study trial mentions the following main objective "Main objective is to select the recommended dose levels of plitidepsin for a future phase II / III efficacy study". However, the authors do not comment at all on their originally intended main objective within the discussion section of their manuscript. The authors need to comment if they were able to determine an optimal dose for future phase II/III trials and mention which dose level would be optimal.

Minor comments:

1. The summary still refers to "effectiveness" rather than "efficacy"

2. The lack of blinding in this study needs to be explicitly mentioned both in the abstract and the manuscript to comply with CONSORT recommendations despite this being a phase I study where absence of blinding is common. Mentioning within the CONSORT checklist that blinding does not apply is not acceptable.

3. Explicitly mention in the abstract how many patients completed the study for every treatment group (1.5, 2.0. and 2.5 mg).

4. Hospital discharge rates cannot be presented in the abstract since they were not pre-specified, unless this clarification is given within the abstract.

5. There is still much confusion around the use of dexamethasone. The first time this is mentioned throughout the manuscript, it suggests that patients were initially not receiving dexamethasone and that dexamethasone was added after a protocol amendment. Please correct this to explicitly mention early in the manuscript if dexamethasone was applied from the beginning to all patients or not and clearer reasons for having made changes in the dose and route of administration particularly for dexamethasone.

6. Why were baseline SpO2 values suppressed from Table 1? This is a very important variable to report, and I can think of no justifications for having deleted it.

7. Table 1 should include the number of patients included per group in the column headings.

8. The full six-point ordinal scale has still not been described in table 1. Please include all categories measured by this scale in the table even though categories like 1 or 4 may be 0 for all patients. It is important to provide the reader all the categories where you intended to classify patients in.

9. Table 2 also needs to include the number of patients for each group alongside the column headings.

10. Hospital discharge cannot be included in table 3 since it was not a pre-specified endpoint, it could be presented in the results section of the manuscript by clarifying that it was not a pre-specified outcome.

11. Figure 4. There seems to be no use in connecting box of cohort five with an arrow to the expansion cohort box. This only adds confusion to the figure. Also, fully describe any abbreviations in the figure legend.

12. Figure 5. There are several problems with this figure. First, time to discharge was not provided as a pre-specified variable to be measured in the study registry nor the study protocol. Second, there is no justification for having deleted log-rank comparisons rather than providing comparisons between all curves as I had previously suggested. Third, it is not clear why the authors are providing comparisons for mild, moderate, and severe patients for a non-pre-specified variable, especially since there is no evident added value of this figure for this particular study; this Kaplan-Meier graph does not really seem to help at all to expand or further clarify the study objectives. I would suggest eliminating this figure from the manuscript unless the authors can convincingly comment why this figure is important for the manuscript and also providing all log-rank comparisons. Similarly, panel B does not really seem to help to further clarify anything according to the study objectives. While panel B can be useful to visualize individual patient trajectories, this figure could be more suitable as supplementary material rather than being one of the main figures. If the authors decide to keep and move this panel to supplementary material, they need to improve visualization since the x-axis have been compressed and they would need to be larger in dimension to be able to properly visualize the graphs. Also, it would be better to have patients with the same dosing next to each other since it is currently too hard to try to figure out why patients with same dosing are dispersed throughout the graphs.

13. Figure S1 does not reflect when amendments to the protocol were made and could make readers think that the intervention was the same throughout the entire study. Please include in this figure all amendments and dates, as well as the number of patients who received each intervention before and after amendments. As I had mentioned before, consider moving this figure to the main manuscript.

14. Change the title of Figure S6 since it could be used to inadequately conclude efficacy. "Inflammatory biomarkers throughout follow-up" could be a more appropriate title.

15. Figure S8 needs to be separated into different figures since readers could inadequately interpret that micrography images

belong to the same patient. Please include data and images from the same patient in one figure and a different figure for murine lung alveoli. Also, a different title is needed to avoid misleading readers. "Comparisons of radiological and inflammatory biomarkers in a 41-year-old patient treated with 2.5 mg/day plitidepsin" could be a better title.

16. Table S6: Eliminate patients discharged from hospital since this was not a pre-specified endpoint as the authors are claiming.

17. It is not correct to say "COVID-19 infection" since COVID-19 is the name of the disease caused by SARS-CoV-2 infection. Please correct throughout the manuscript. Also, the correct name is "coronavirus disease (COVID-19)", not "coronavirus disease 2019 (COVID-19)". See the following link for clarifications: [https://www.who.int/emergencies/diseases/novel-coronavirus-2019/technical-guidance/naming-the-coronavirus-disease-\(covid-2019\)-and-the-virus-that-causes-it](https://www.who.int/emergencies/diseases/novel-coronavirus-2019/technical-guidance/naming-the-coronavirus-disease-(covid-2019)-and-the-virus-that-causes-it)

18. The statistical analysis section mentions that regression analyses were done, however, the authors declared in their rebuttal that they had decided to suppress regression analyses. Please correct accordingly.

19. In the phrase "While on study, 64.4% (29 of 45) of patients received systemic corticosteroids, beyond their use as pre-medication", please mention the mean duration of treatment for these patients and the reason(s) why systemic corticosteroids were extended.

20. It may not be appropriate to refer to "rapid decline" in the results section since this adjective can be quite ambiguous and misleading. Instead, it may be better to describe the results and allow the readers to interpret by themselves the results.

21. The following phrase cannot be in the results section since it carries an interpretation of data. Interpretations need to be left for the discussion section: "These findings may reflect immunomodulatory/anti-inflammatory effects that could be either secondary to the reduction in viral load or mediated by plitidepsin, and might contribute to the rapid recovery of lung infiltrates reported in some patients with computed tomography scan evaluations".

Lastly, the authors did not include track changes for the current revised manuscript, which diffculted reviewing the manuscript. Please highlight all changes for the re-submission of the manuscript.

Reviewer #3 (Comments to the Authors (Required)):

The authors performed a great improvement of their manuscript by answering to the reviewer's comments and toned down their claims in order to conclude on their results. Figures are clearer and the authors do not perform over interpretation. The title is a proper statement and I recommend the paper for acceptance.

José F. Varona MD
Hospital Universitario
HM Montepíncipe (Madrid)
Department of Internal Medicine
E-mail: jfvarona@hmhospitales.com
December 14, 2021

Eric Sawey, PhD
Life Science Alliance
Executive Editor

Dear Dr. Sawey,

On behalf of the authors, I would like to thank you again for your continued consideration of our manuscript titled “Pre-clinical and randomized phase I studies of plitidepsin in adults hospitalized with COVID-19 ” (LSA-2021-01200-T).

We understand that the general policy of *Life Science Alliance* is to only consider papers through a single revision cycle, and therefore are grateful for the opportunity to revise our manuscript a second time.

Below, we address the reviewers’ comments point by point.

Reviewer 2

As before, we thank the reviewer for their comprehensive review of our manuscript. Below, we address the reviewer’s concerns point by point.

- *In this resubmission, the authors have addressed some of the main comments I had made on their first version of their manuscript, while also making significant efforts to adequately report their manuscript according to CONSORT recommendations. However, they did not adequately address all reviewers' main comments. Also, unsolicited modifications were made, and new results have been added without providing any explanations, as well as the inclusion of one more author (which I am unsure if is allowed by the journal). Therefore, this resubmission required a full re-evaluation and major and minor revisions are needed.*

We would like to apologize for involuntarily neglecting some of the previous comments from this reviewer. We have been especially careful during this second review to ensure that we have addressed every point that has been raised.

As the medical community is aware, the ongoing pandemic is a dynamic situation, with, for instance, new variants and treatment modalities constantly emerging. In this regard, we have included the latest results of ongoing preclinical research to this most recent draft of the manuscript, adding new and relevant information on the activity of plitidepsin across additional SARS-CoV-2 variants. This new data has required the incorporation of two additional researchers to the author list from Dr Adolfo Garcia-Sastre’s lab (Icahn School of Medicine at Mount Sinai, New York, NY): Briana L. McGovern and M. Luis Rodriguez, both of whom were responsible for the research experiments performed in biosafety level 3 (BSL-3) facilities.

Additionally, we would like to clarify that the previous addition of A. Belgrano (statistician), which was spotted by the reviewer, was due an involuntary omission in the first draft, which was transferred from another journal.

- *“On the first version of the manuscript, both reviewers argued that efficacy could not be concluded from the authors' results. The authors have not properly addressed our comments. While they mention that they now focus more on safety rather than efficacy endpoints, they still make claims of efficacy and there is no contextualization of their results according to the natural course of the disease as I had previously commented on; it feels like our arguments were simply ignored. For instance, the following sentences are evidence of the authors claiming efficacy which is not supported by their study: "Patients treated with plitidepsin showed rapid reduction in viral load (compared to their baseline value), analytical improvement of biomarkers associated to inflammatory processes (Figures S4 and S5), and there is evidence of prompt clearance of pneumonia infiltrates in some participants with available chest imaging performed for medical reasons (i.e. not per protocol) (Figure S8A-B). Each of these outcomes very likely contributed to mitigating disease progression and leading to an earlier discharge from the hospital". Please delete all claims of efficacy throughout the manuscript since they are not supported by this study.”*

Again, we agree with the reviewer that this study was primarily aimed to evaluate the safety and tolerability of three doses of plitidepsin in hospitalized patients with COVID-19. Nevertheless, it was also designed to evaluate, as secondary endpoints, potential measures of efficacy including change from baseline in viral load and some clinical outcomes, such as death rate and increasing needs for respiratory support. Therefore, we have reported these preliminary efficacy data, though we have been careful to note, in various locations throughout the manuscript, that a causal link between plitidepsin treatment and these secondary endpoints cannot be established without a placebo-controlled clinical trial. We have also explicitly mentioned if the results came from *post-hoc* analyses.

- *While it is appropriate to report all outcomes as originally described in the study registry and study protocol, the authors must make a greater effort to fully explain the meaning of their results by commenting the following points in the discussion section, which were also left ignored in their revision:
Several studies have shown that viral load peaks at symptom onset with subsequent decline in viral load. This is due to the natural course of the disease, reason why viral load decline in this study cannot be used to conclude preliminary efficacy since there was no control group. References:
Néant N, Lingas G, Le Hingrat Q, Ghosn J, Engelmann I, Lepiller Q, et al. Modeling SARS-CoV-2 viral kinetics and association with mortality in hospitalized patients from the French COVID cohort. *Proc Natl Acad Sci.* 2021 Feb 23;118(8):e2017962118.
He X, Lau EHY, Wu P, Deng X, Wang J, Hao X, et al. Temporal dynamics in viral shedding and transmissibility of COVID-19. *Nat Med.* 2020 May 15;26(5):672-5.
For the same reason, negativization of RT-PCR cannot be used to conclude preliminary efficacy in this study in the absence of a control group.*

We thank the reviewer for these comments and have included a sentence in the discussion on the natural progression of COVID-19 viral load. This new sentence helps support the subsequent line in the same paragraph, which highlights the need for a control group in future studies.

- *This study was not suitable to evaluate preliminary efficacy on mortality nor oxygen therapy modalities since a control group was not included. Strict inclusion and exclusion criteria due to this being a phase I study limit the authors' ability to compare their findings to what has been reported in the literature. The authors failed to acknowledge these points in the limitations of their study.*

We thank the reviewer for this comment. We had thought our discussion of the study limitations to be sufficient and concise, but have expanded the text to highlight that the key study limitations (small number of patients, high observed variability, and lack of a control group) prevent conclusions of efficacy, limit our ability to observe dose-response effects, and restrict our use of this data to only hypothesis generation.

- *The authors mention in the manuscript that hospital discharge was a secondary outcome, however this endpoint was not described in the trial registry, nor the trial protocol provided. It is therefore not acceptable to present their results in terms of hospital discharge rates. Outcomes need to be presented exactly as described in the study registry and protocol throughout the entire manuscript, including the abstract. All results of discharge rates could be moved to the post-hoc analysis section while also commenting in the discussion section that discharge rates were not a pre-defined endpoint which warrants caution for the interpretation of these results.*

We have removed hospital discharge from the abstract, and revised the body of the manuscript such that discharge rates are presented in a sub-section on *post-hoc* outcome analyses. We have also specified in the Discussion that, "... hospital discharge rates, changes in inflammatory biomarkers, and radiological studies were not pre-defined endpoints in this trial. Thus, this study was not designed to evaluate if plitidepsin can improve these parameters and ongoing and future controlled clinical studies will address these hypotheses."

- *Improvement of inflammatory biomarkers and radiological improvements were also not primary, secondary, or exploratory endpoints. Thus, the authors cannot use these results to argue efficacy since the trial was not designed to address these hypotheses. All comments on improvements in radiological and inflammation parameters need to be omitted in the discussion section, otherwise the authors need to include the following statement within the discussion: "Changes in inflammatory biomarkers and radiological studies were not pre-defined endpoints in this trial. Thus, this study was not designed to evaluate if plitidepsin can improve these parameters and future studies could be designed to address these hypotheses".*

We thank the reviewer for this suggestion, and have incorporated their recommended wording into the discussion section of the revised manuscript, including hospital discharge rates as noted above.

- *Version 7.0 of the trial protocol has been provided by the authors as requested. However, protocol amendments for all prior versions are not available for scrutiny. These need to be provided within the protocol as a subheading. See SPIRIT recommendations (specifically item 25) which explain why key protocol amendments for all versions need to be described: <https://www.spirit-statement.org/protocol-amendments/>*

We agree with the reviewer that for full transparency, all protocol amendments should be provided. We provide them in Supplement 3 along with a methodological description of the *post-hoc* analyses that were conducted.

- *The administration of other drugs (dexamethasone, diphenhydramine, ranitidine, and ondansetron) could be considered a major deviation from the international study registry since the use of these pharmaceuticals was unfortunately not described at all at any moment within the ClinicalTrials.gov registry. Since the authors have not provided information on protocol version amendments, it is impossible to corroborate exactly when and why co-interventions were implemented, which is one of the main reasons why providing all amendments to the protocol is very important for this study.*

We thank the reviewer for bringing this to our attention. As noted above, a summary of all the amendments has been added to the manuscript as Supplement 3. Furthermore, we have initiated the process of having the trial registry revised to include the protocol-defined pre-medication.

- *Moving Supplementary Figure 1 to the main manuscript would also allow readers to have a clearer view of how the intervention was applied. Please consider having this figure in the main manuscript instead of supplementary material.*

We agree with this comment and have followed this useful suggestion to increase the clarity of the revised manuscript.

- *I had asked the authors to include in the discussion of their manuscript the potential implications of having combined dexamethasone with plitidepsin, which was also ignored. Namely, they should discuss (explicitly in the discussion section of the manuscript): 1) Acknowledge that dexamethasone may be a major confounder since it appears to be the case that all patients received plitidepsin alongside dexamethasone as the authors declared in their rebuttal. 2) In order to address this confounding, what kind of trial designs will be needed in the future (i.e., factorial randomized controlled trials)? This is important to discuss since we need to understand what may really work on patients, not only seek to approve drug combinations without knowing which components may or may not work. 3. What kind of patients (according to disease severity) could likely benefit from the combination of plitidepsin and dexamethasone and could some patients be harmed (i.e., mild COVID-19)? See the following reference for any clarifications of what I meant in the first round of review: Rochwerf B, Agarwal A, Siemieniuk R A, Agoritsas T, Lamontagne F, Askie L et al. A living WHO guideline on drugs for covid-19 BMJ 2020; 370 :m3379 doi:10.1136/bmj.m3379. 4) If the authors consider that dexamethasone is not a major confounder, well-founded counter-arguments need to be provided in the discussion section to convincingly clarify why it is not a confounder since only mentioning that the authors do not think that dexamethasone is a confounder is not acceptable. 5) There is evidence that dexamethasone can delay viral clearance in patients with mild disease (<http://www.pnas.org/lookup/doi/10.1073/pnas.2017962118>), how could this affect patients receiving dexamethasone plus plitidepsin? If the answer is "I am not sure", then this needs to be recognized in the discussion as a potential risk until studies are performed to evaluate this hypothesis. It is important to discuss all these points so that this paper can be useful to continue investigating drugs for COVID-19 in a safe way while considering all potential risks.*

We thank the reviewer for this comment, and did not mean to give the appearance that we had ignored their previous request. In addition to improving the description of the premedication regimen and its modification, we have added a new paragraph to the discussion that addresses each of the points raised here, and includes information that we had previously only shared with the reviewers.

- *The mean 15-day viral load reduction value has been changed in the abstract without any explanations.*

We would like to apologize for this oversight. While addressing the previous reviewer comments regarding patients with undetectable baseline viral load (comment 16 in the Results section from Reviewer 2) we went back to the lead scientist of SynLab, the vendor which performed the qt-PCR analyses for this study. They clarified to us that the lower limit of quantitation was not 10 copies/mL, as stated previously, but 10 mL/run, which translates into 1000 copies/mL. Therefore, all values previously listed as zero were, in fact, undetectable values. We subsequently adopted the consensus of imputing an intermediate value between 0-3 (log₁₀) to samples with undetectable/unquantifiable

results. This was explained in the methods section, but involuntarily neglected to mention it in our response to the reviewers.

- *Also, it is not clear why the only secondary outcome being presented is the result for viral clearance at day 15 since this outcome was set to be measured up to day 31.*

We thank the reviewer for this observation, and note that we now report all available values to day 31.

- *The authors should either describe all secondary outcomes and present all results for secondary outcomes in the abstract, or leave all secondary outcomes for the main manuscript. Notice that CONSORT only requires the primary outcome to be presented from the abstract.*

We have chosen to report all of the secondary outcomes in the abstract, in the order that they are presented in the study protocol. We note that because of this addition, the abstract is slightly over the word count for articles published in *Life Science Alliance*, and would request that editor grant us an exception in order to fully present our data.

- *The study trial mentions the following main objective "Main objective is to select the recommended dose levels of plitidepsin for a future phase II / III efficacy study". However, the authors do not comment at all on their originally intended main objective within the discussion section of their manuscript. The authors need to comment if they were able to determine an optimal dose for future phase II/III trials and mention which dose level would be optimal.*

We thank the reviewer for this comment. They are correct that results from the trial did not enable us to identify the plitidepsin dose associated with the most favorable risk-benefit ratio, and therefore the lowest and the highest doses were selected for comparison in a phase 3 setting against a control arm. This has been explicitly noted in the discussion (page 11-line 25 and page 12-penultimate paragraph).

- *Minor comments:
The summary still refers to "effectiveness" rather than "efficacy"*

We have changed the phrase from "...may be effective..." to "...may be efficacious..."

- *The lack of blinding in this study needs to be explicitly mentioned both in the abstract and the manuscript to comply with CONSORT recommendations despite this being a phase I study where absence of blinding is common. Mentioning within the CONSORT checklist that blinding does not apply is not acceptable.*

We thank the reviewer for this clarification, and have added the term 'open-label' to the study description in the abstract and materials & methods.

- *Explicitly mention in the abstract how many patients completed the study for every treatment group (1.5, 2.0. and 2.5 mg).*

We have added to the abstract the numbers of patients who were included in each cohort and how many completed treatment.

- *Hospital discharge rates cannot be presented in the abstract since they were not pre-specified, unless this clarification is given within the abstract.*

Although we found it both useful and clinically relevant to show this information, we agree that formally speaking this represents the result from a *post-hoc* analysis, and therefore we will not present it in the abstract.

- *There is still much confusion around the use of dexamethasone. The first time this is mentioned throughout the manuscript, it suggests that patients were initially not receiving dexamethasone and that dexamethasone was added after a protocol amendment. Please correct this to explicitly mention early in the manuscript if dexamethasone was applied from the beginning to all patients or not and clearer reasons for having made changes in the dose and route of administration particularly for dexamethasone.*

As noted above, we have tried to make this important information clearer throughout the manuscript. Please see Figure 5 as well as descriptions in the Results, Discussion, and Supplement 3.

- *Why were baseline SpO2 values suppressed from Table 1? This is a very important variable to report, and I can think of no justifications for having deleted it.*

During the previous revision we chose to remove these data because they represented the values of the 22 patients whose oxygen saturation was assessed at room air conditions. and therefore this could be misleading information. But the reviewer is correct that this could be important to clinicians, and we have added these data back, with additional descriptions for a better understanding of the type of patients who entered into the study

- *Table 1 should include the number of patients included per group in the column headings.*

Numbers of patients per group have been added to the table.

- *The full six-point ordinal scale has still not been described in table 1. Please include all categories measured by this scale in the table even though categories like 1 or 4 may be 0 for all patients. It is important to provide the reader all the categories where you intended to classify patients in.*

We appreciate the reviewer's concern, but now that the numbers of patients have been added to the column headings, we believe it is obvious that all patients are accounted for in the two categories that are filled. Adding rows of zeroes for the other four categories may be unnecessarily complicating. Nevertheless, a full description of the 6-point ordinal scale (including each of the categories, even those that are not shown) has been added to the footnote of the table and to Supplement 3.

- *Table 2 also needs to include the number of patients for each group alongside the column headings.*

We have moved the numbers of patients from the table footnote to the column headings.

- *Hospital discharge cannot be included in table 3 since it was not a pre-specified endpoint, it could be presented in the results section of the manuscript by clarifying that it was not a pre-specified outcome.*

We have removed hospital discharge from table 3, leaving only a description of it in the body of the manuscript, under *post-hoc* analyses.

- *Figure 4. There seems to be no use in connecting box of cohort five with an arrow to the expansion cohort box. This only adds confusion to the figure. Also, fully describe any abbreviations in the figure legend.*

We thank the reviewer for this suggestion and have removed the arrow between cohort five and the expansion cohort. We have also added the abbreviations to the figure legend.

- *Figure 5. There are several problems with this figure. First, time to discharge was not provided as a pre-specified variable to be measured in the study registry nor the study protocol. Second, there is no justification for having deleted log-rank comparisons rather than providing comparisons between all curves as I had previously suggested. Third, it is not clear why the authors are providing comparisons for mild, moderate, and severe patients for a non-pre-specified variable, especially since there is no evident added value of this figure for this particular study; this Kaplan-Meier graph does not really seem to help at all to expand or further clarify the study objectives. I would suggest eliminating this figure from the manuscript unless the authors can convincingly comment why this figure is important for the manuscript and also providing all log-rank comparisons. Similarly, panel B does not really seem to help to further clarify anything according to the study objectives. While panel B can be useful to visualize individual patient trajectories, this figure could be more suitable as supplementary material rather than being one of the main figures. If the authors decide to keep and move this panel to supplementary material, they need to improve visualization since the x-axis have been compressed and they would need to be larger in dimension to be able to properly visualize the graphs. Also, it would be better to have patients with the same dosing next to each other since it is currently too hard to try to figure out why patients with same dosing are dispersed throughout the graphs.*

As noted above, we have clarified throughout the manuscript that time to discharge was a *post-hoc* analysis. This is now also stated in the title of the figure. The original figure was comprised of two panels. To aid in visualization, we have elected to separate them into two separate figures, which have been moved to the supplement as Figures S3 and S4.

Our rationale for including the analysis of hospital discharge by disease severity is simply hypothesis generation for future studies, including the currently ongoing phase III trial NCT04784559, though we agree that this is not the type of analysis that can be directly derived from trial design. Nevertheless it provides clues for future development of plitidepsin in COVID-19, including the proposal of target population for clinical research. However, following the reviewer's comments and concerns, in the current version of the manuscript, we have moved this analysis to the supplement (Figure S3), but have replaced it with a figure showing hospital discharge by plitidepsin dose (Figure 7 panel A).

With regard to the statistical analysis for the reverse Kaplan Meier graphs, we performed the log-rank test with the null hypothesis that each of the curves was equal. As it was rejected, the conclusion is that at least one curve is different. To ascertain whether all these curves are different from each other, one approach is to perform pairwise comparisons between group levels with corrections for multiple testing. Results of the pairwise comparisons are now shown in Figures 6 and S6.

As mentioned here, panel B has now become Figure S4 for better visualization. We have grouped patients with the same dosing level, which are represented in Figure 7 panel B.

- *Figure S1 does not reflect when amendments to the protocol were made and could make readers think that the intervention was the same throughout the entire study. Please*

include in this figure all amendments and dates, as well as the number of patients who received each intervention before and after amendments. As I had mentioned before, consider moving this figure to the main manuscript.

We thank the reviewer for this comment and have moved this figure to the main manuscript (Figure 5). Additionally, we have added a new panel clearly listing all of the amendments affecting the interventions of the study along with the dates that they were changed.

- *Change the title of Figure S6 since it could be used to inadequately conclude efficacy. "Inflammatory biomarkers throughout follow-up" could be a more appropriate title.*

We thank the reviewer for this suggestion and have made the change accordingly.

- *Figure S8 needs to be separated into different figures since readers could inadequately interpret that micrography images belong to the same patient. Please include data and images from the same patient in one figure and a different figure for murine lung alveoli. Also, a different title is needed to avoid misleading readers. "Comparisons of radiological and inflammatory biomarkers in a 41-year-old patient treated with 2.5 mg/day plitidepsin" could be a better title.*

We thank the reviewer for this suggestion and have both separated the figures and adapted the titles as recommended.

- *16. Table S6: Eliminate patients discharged from hospital since this was not a pre-specified endpoint as the authors are claiming.*

We have deleted the rows on hospital discharge.

- *17. It is not correct to say "COVID-19 infection" since COVID-19 is the name of the disease caused by SARS-CoV-2 infection. Please correct throughout the manuscript. Also, the correct name is "coronavirus disease (COVID-19)", not "coronavirus disease 2019 (COVID-19)". See the following link for clarifications: [https://www.who.int/emergencies/diseases/novel-coronavirus-2019/technical-guidance/naming-the-coronavirus-disease-\(covid-2019\)-and-the-virus-that-causes-it](https://www.who.int/emergencies/diseases/novel-coronavirus-2019/technical-guidance/naming-the-coronavirus-disease-(covid-2019)-and-the-virus-that-causes-it)*

We thank the reviewer for this clarification and have corrected the terminology throughout the manuscript.

- *18. The statistical analysis section mentions that regression analyses were done, however, the authors declared in their rebuttal that they had decided to suppress regression analyses. Please correct accordingly.*

We apologize for the misunderstanding, but, as per the reviewer's previous comment (point 10 in the Results section), we had only removed the regression analysis used for predicting the likelihood of patients being discharged and alive at Day 8. Other regression analyses are still valid, such as those used in the *post-hoc* exploratory analysis of the correlation between viral load and hospital discharge at Day 15.

- *In the phrase "While on study, 64.4% (29 of 45) of patients received systemic corticosteroids, beyond their use as pre-medication", please mention the mean duration of treatment for these patients and the reason(s) why systemic corticosteroids were extended.*

As previously mentioned, we have improved the description of the use of corticosteroids throughout the manuscript.

- *It may not be appropriate to refer to "rapid decline" in the results section since this adjective can be quite ambiguous and misleading. Instead, it may be better to describe the results and allow the readers to interpret by themselves the results.*

We have removed the adjective ‘rapid’ from all places that it appeared in the Results section. However, we have left it in the discussion where we provide our interpretation (see next point).

- *The following phrase cannot be in the results section since it carries an interpretation of data. Interpretations need to be left for the discussion section: "These findings may reflect immunomodulatory/anti-inflammatory effects that could be either secondary to the reduction in viral load or mediated by plitidepsin, and might contribute to the rapid recovery of lung infiltrates reported in some patients with computed tomography scan evaluations".*

We have revised the sentence so as not to include any interpretation, and is only a statement of the improvement of lung infiltrates that was observed.

- *Lastly, the authors did not include track changes for the current revised manuscript, which diffculted reviewing the manuscript. Please highlight all changes for the re-submission of the manuscript.*

We have ensured that all additional changes have been captured with the ‘track change’ feature turned on.

Reviewer #3 (Comments to the Authors (Required)):

- *The authors performed a great improvement of their manuscript by answering to the reviewer's comments and toned down their claims in order to conclude on their results. Figures are clearer and the authors do not perform over interpretation. The title is a proper statement and I recommend the paper for acceptance.*

We again thank the reviewer for their previous comments, which have greatly improved the manuscript.

We are again grateful at the opportunity to address the reviewers' comments as they have further improved our original manuscript. We hope that our revisions and responses have been satisfactory to assuage their concerns.

As always, we are happy to address any other questions or comments that arise after this most recent revision. Please, do not hesitate to contact us if there is anything else we can do.

Sincerely,

José F. Varona MD
Hospital Universitario HM Montepíncipe (Madrid)
Department of Internal Medicine
Corresponding author.
E-mail: jfvarona@hmhospitales.com.

December 20, 2021

RE: Life Science Alliance Manuscript #LSA-2021-01200-TRR

Dr. Jose F Varona
Departamento de Medicina Interna
Hospital Universitario HM Montepíncipe
HM Hospitales
Madrid, Spain

Dear Dr. Varona,

Thank you for submitting your revised manuscript entitled "Pre-clinical and randomized phase I studies of plitidepsin in adults hospitalized with COVID-19". We would be happy to publish your paper in Life Science Alliance pending final revisions necessary to meet our formatting guidelines.

- please add the Twitter handle of your host institute/organization as well as your own or/and one of the authors in our system
- please consult our manuscript preparation guidelines <https://www.life-science-alliance.org/manuscript-prep> and make sure your manuscript sections are in the correct order
- please upload your Tables in editable .doc or excel format
- Tables can be included at the bottom of the main manuscript file or be sent as separate files.
- please add your main, supplementary figure, and table legends to the main manuscript text after the references section
- Please upload all figure files as individual ones, including the supplementary figure files; all figure legends should only appear in the main manuscript file
- please add scale bars to Figure S10, and indicate their size in the figure legend

A. FINAL FILES:

B. MANUSCRIPT ORGANIZATION AND FORMATTING:

Sincerely,

December 28, 2021

RE: Life Science Alliance Manuscript #LSA-2021-01200-TRRR

Dr. Jose F Varona
Departamento de Medicina Interna
Hospital Universitario HM Montepíncipe
HM Hospitales
Madrid, Spain

Dear Dr. Varona,

Thank you for submitting your Research Article entitled "Pre-clinical and randomized phase I studies of plitidepsin in adults hospitalized with COVID-19". It is a pleasure to let you know that your manuscript is now accepted for publication in Life Science Alliance. Congratulations on this interesting work.

DISTRIBUTION OF MATERIALS:

Again, congratulations on a very nice paper. I hope you found the review process to be constructive and are pleased with how the manuscript was handled editorially. We look forward to future exciting submissions from your group.

Sincerely,
